

# Measurement Report: Seasonal variation and anthropogenic influence on cloud condensation nuclei (CCN) activity in the South China Sea: Insights from shipborne observations during summer and winter of 2021

Hengjia Ou[1], Mingfu Cai[2], Yongyun Zhang[1], Xue Ni[1], Baoling Liang[3], Qibin Sun[4,5],

Shixin Mai[1], Cuizhi Sun[6], Shengzhen Zhou[1], Haichao Wang[1], Jiaren Sun[2], Jun Zhao[1]

[1]School of Atmospheric Sciences, Guangdong Province Key Laboratory for Climate Change and Natural

Disaster Studies, Southern Marine Science and Engineering Guangdong Laboratory (Zhuhai), Sun Yat-

sen University, Zhuhai, Guangdong 519082, China

[2]Guangdong Province Engineering Laboratory for Air Pollution Control, Guangdong Provincial Key

Laboratory of Water and Air Pollution Control, South China Institute of Environmental Sciences, MEE,

Guangzhou 510655, China

[3]Guangzhou Sub-branch of Guangdong Ecological and Environmental Monitoring Center, Guangzhou

510006, China

[4]Dongguan Meteorological Bureau, Dongguan, Guangdong, 523086, China

[5]Dongguan Engineering Technology Research Center of Urban Eco-Environmental Meteorology,

Dongguan, Guangdong, 523086, China

[6]Southern Marine Science and Engineering Guangdong Laboratory (Zhuhai), Zhuhai, Guangdong

519082, China

*Correspondence:* Mingfu Cai (caimingfu@scies.org) and Jun Zhao (zhaojun23@mail.sysu.edu.cn)



**Abstract**
Understanding seasonal variation in cloud condensation nuclei (CCN) activity and the impact of
anthropogenic emissions in marine environments is crucial for assessing climate change. In this study,
two shipborne observations in the South China Sea (SCS) during the summer and winter of 2021 were
conducted. During summer, higher particle number concentrations but lower mass concentrations of non-
refractory submicron particles (NR-PM$_1$) were observed. These differences were attributed to the
dominance of particles in the Aitken mode during summer and in the accumulation mode during winter.
Moreover, particles during summer were more hygroscopic with higher activation ratios (ARs) at all
supersaturation (SS). Based on backward trajectory analysis, the whole campaign was classified into
terrestrial and mixed air mass influence periods. Particles measured during the terrestrial period
consistently exhibited lower hygroscopicity values. Additionally, minor variations were shown for all
NR-PM$_1$ components under different air mass influences during summer, while the mass fraction of
nitrate increased significantly under terrestrial influence during winter. Particle number size distribution
(PNSD) exhibited unimodal distribution during terrestrial period and bimodal distribution during mixed
air mass influence period, with winter displaying a more pronounced bimodal pattern than summer. The
impact of PNSD on AR was greater than on aerosol hygroscopicity in summer, and vice versa in winter.
During terrestrial period, significant variations in PNSD were observed with the offshore distance, and
the largest variation was seen in Aitken mode during both summer and winter. Meanwhile, aerosol
hygroscopicity shows an increasing trend with the offshore distance, which is primarily attributed to the
increase of sulfate fraction during summer and the decrease of the black carbon fraction during winter.
Using a single parameterized PNSD in the $N_{CCN}$ prediction can lead to errors exceeding 100% during
both summer and winter, with dominant terrestrial air masses in the SCS atmosphere, while using a
constant hygroscopicity parameter would lower the errors in the $N_{CCN}$ prediction (~15% during winter
and ~10% during summer). Our study shows significant differences in aerosol properties between winter
and summer seasons and highlights the influence of anthropogenic emissions on the CCN activity in the
SCS.



**1.Introduction**
Aerosols can act as cloud condensation nuclei (CCN), influencing cloud formation, lifespan, and
albedo, thus indirectly impacting global radiative balance (Fletcher et al., 2011; Albrecht, 1989). The
aerosol-cloud interaction currently represents the largest uncertainty in radiative forcing within climate
models, ranging from -1.7 to -0.3 W m$^{-2}$ (IPCC, 2021). This uncertainty can be attributed to the significant
spatiotemporal variability in the aerosol size distribution and the ability of atmospheric aerosol particles
acting as CCN (CCN activity) (Fitzgerald, 1973). Thus, field measurements of aerosol size distribution
and physicochemical properties are needed to better understand the radiative forcing exerted by
atmospheric aerosol particles.
Previous studies suggest that particle number size distribution (PNSD) is a primary factor
influencing CCN concentrations (Dusek et al., 2006; Rose et al., 2010; Pöhlker et al., 2016; Burkart et
al., 2011). The PNSD can account for 84–96% of the variability (Dusek et al. (2006) in CCN
concentrations ($N_{CCN}$), while CCN activities may also play a significant role in CCN concentrations
(Quinn et al., 2008; Cai et al., 2018; Ovadnevaite et al., 2017; Liu et al., 2018; Crosbie et al., 2015),
which are primarily governed by the particle size, chemical composition, surface tension, and
hygroscopicity (Köhler, 1936; Seinfeld and Pandis, 2016). Among these factors, the impact of
hygroscopicity on CCN activities has received great attention in recent years (Petters and Kreidenweis,
2007; Ajith et al., 2022; Rose et al., 2010). Petters and Kreidenweis (2007) proposed the κ- Köhler theory
based on the Köhler theory to quantify the ability of aerosol particles to absorb moisture and become
CCN based on the aerosol hygroscopicity parameters (κ). Ajith et al. (2022) showed that 64% of particles
can be activated as CCN when κ is equal to 0.37, whereas when κ decreases to 0.23, only 48% of particles
can be activated.
Previous field observations have indicated significant seasonal variations in PNSD and
hygroscopicity under both terrestrial and marine environments, leading to the seasonal variations in $N_{CCN}$
(Crosbie et al., 2015; Schmale et al., 2018; Burkart et al., 2011; Bougiatioti et al., 2009; Sihto et al., 2011;
Leena et al., 2016; Ross et al., 2003; Gras and Keywood, 2017; Quinn et al., 2019). Crosbie et al. (2015)
revealed that in the urban area of Arizona particles had larger sizes, higher hygroscopicity, and $N_{CCN}$ was
also higher during winter, while a higher abundance of smaller particles was observed during summer
owing to stronger photochemical reactions. In pristine environments like mountain, coastal, and forested



regions, seasonal variations in $N_{CCN}$ and PNSD were more pronounced than urban and rural areas
(Schmale et al., 2018). Pöhlker et al. (2016) observed significant differences in $N_{CCN}$ between the wet
and dry seasons in the Amazon rainforest, while the κ values remained relatively stable. They also noted
increased particle concentrations and aerosol hygroscopicity, both subject to the impact of long-range
transport originating from anthropogenic emissions. Observations in marine areas during different
seasons are relatively scarce compared with those in inland areas. Gras (1995) found that both particle
concentration and $N_{CCN}$ in the Southern Ocean reached their peaks during summer and gradually decrease
to their valleys in winter. Quinn et al. (2019) showed that sea spray aerosols make a relatively significant
contribution to $N_{CCN}$ only during winter in the Western North Atlantic, while in other seasons, the primary
contribution comes from biogenic aerosols oxidized from dimethyl sulfide (DMS). Zheng et al. (2020)
revealed that sulfate dominates the particle condensational growth to CCN sizes during summer in the
North Atlantic, while secondary organic aerosols played a significant role in particle growth throughout
all seasons. These results indicate that CCN activity and concentration could vary in a large range during
different seasons. Thus, further observations across different seasons in marine environments are needed
to enhance our understanding of marine CCN activities and their seasonal variations.
The South China Sea (SCS) is located between the Asian continent to the north and the Indonesian
archipelago to the south. The aerosols in the SCS can be influenced by ship emissions, marine
background emissions, and transport of terrestrial air masses from China, Indochinese Peninsula, and
Philippines, making their sources relatively complex (Geng et al., 2019; Liang et al., 2021; Sun et al.,
2023). The complex aerosol sources in the SCS lead to the complexity of PNSD and CCN activity in this
area. Atwood et al. (2017) found that under the influence of marine air masses, particles exhibited a
significant bimodal distribution with a κ of 0.65, whereas under the influence of continental air masses,
a unimodal distribution and a κ of 0.4 were observed. Another summer study in the northern SCS revealed
a predominantly unimodal aerosol PNSD with a peak centering at 60–80 nm and an average κ value of
0.4 (Cai et al., 2020). Moreover, particle and $N_{CCN}$ decreased with increasing offshore distance,
highlighting the significant influence of mainland China on CCN activity in the northern SCS (Cai et al.,
2020). However, our understanding on CCN activity, particularly the variations in CCN activity in the
SCS across different seasons, remains largely unknown. Conducting observational studies on CCN
activity across different seasons would promote our understanding of CCN activity and, ultimately,
reduce uncertainties related to aerosol-cloud interactions and radiative forcing in the SCS.



In this study, we conducted two shipborne observations in the SCS during summer (May 5–June 9,
2021) and winter (December 19–29, 2021). Our observations with online instruments focused on
measuring aerosol chemical composition, PNSD, and CCN activation in the region. Our results provide
valuable insights into the differences in CCN activity between winter and summer, as well as the
influence of terrestrial transport on CCN activity in the SCS across different seasons.
**2. Methodology**
**2.1 Cruise information and onboard measurements**
**2.1.1 Cruise information**
This study consists of two research cruises conducted during the summer and winter of 2021,
respectively. The summer and winter cruises were carried out respectively by the vessels "Tan Kah Kee"
and "Sun Yat-sen University". The "Tan Kah Kee" is an oceanographic research vessel with a length of
77.7 meters, a beam of 16.24 meters, and a displacement of 3611 tons. The "Sun Yat-sen University" is
a comprehensive oceanographic training vessel with a total length of 114.3 meters, a beam of 19.4 meters,
and a displacement of 6880 tons.
The first cruise was from May 5th to June 9th, 2021.  The cruise started from Xiamen Port and
traversed from the northern to the central-southern South China Sea, and then circled back near Hainan
Island, and finally returned to Xiamen Port. The second cruise was from December 19th to December
29th, 2021. It began from Gaolan Port in Zhuhai and reached the vicinity of Yongxing Island, and
ultimately returned to Gaolan Port (Fig. 1). On both cruises, most of the instruments were housed in a
single compartment and the sampling lines were extended from the window of the compartment to the
height of the ship's bridge (Fig. S1).
**2.1.2 Size-resolved cloud condensation nuclei activity measurement**
The size-resolved CCN activity was measured with a combination of a scanning mobility particle
sizer (SMPS) system and a cloud condensation nuclei counter (model CCNc-200, DMT Inc., USA), the
SMCA method initially proposed in Moore et al. (2010). The SMPS system consisted of a differential
mobility analyzer (DMA; model 3082, TSI., Inc.) and a condensation particle counter (CPC; model 3756,
TSI Inc.). The SMPS and the CCNc system were set to measure PNSD and size-resolved CCN number



137 concentration at a mobility size range of 10–500 nm and 10–593 nm in summer and winter campaign,

138 respectively.

139   The supersaturation (SS) of the CCNc was set at 0.2 %, 0.4 %, and 0.7 % in summer campaign and

140 0.1%, 0.2 %, 0.4 %, and 0.7 % in winter campaign, respectively. Before the measurements, the CCNc

141 was calibrated with ammonium sulfate ($(NH_4)_2SO_4$) particles at each set SS. Detailed description of the

142 instrument configuration and calibration can be found in Cai et al. (2018).

### 2.1.3 Aerosol chemical composition measurement

144   The chemical composition of atmospheric non-refractory submicron particulate matter (NR-PM$_1$),

145 including sulfate, nitrate, organics, ammonium, and chloride, was measured using an online time-of-

146 flight ACSM (ToF-ACSM; Aerodyne Inc., USA). The sampling time of the ToF-ACSM was

147 approximately 10 min. The relative ionization efficiency (RIE) values of the instrument were calibrated

148 using ammonium nitrate ($NH_4NO_3$) and ammonium sulfate ($(NH_4)_2SO_4$) both before the start and after

149 the completion of the campaigns. The RIE values for ammonium were 3.31 and 3.33 during the summer

150 and winter, respectively, while the ones for sulfate were 1.02 and 0.81 during the summer and winter,

151 respectively. The collection efficiency (CE) was determined as shown in Sun et al. (2023) and time-

152 independent CE values were used in this study. Detailed CE calculation can be found in the

153 supplementary (Text S2). The organic carbon (OC)/elemental carbon (EC) concentrations in PM$_{2.5}$ were

154 measured using a semi-continuous OC/EC analyzer (Model-4, Sunset Laboratory Inc., USA) based on

155 the thermal optical transmittance technique and detailed measurement process can be found in Sun et al.

156 (2023). The black carbon concentrations were measured with an aethalometer (AE33, Magee Scientific).

### 2.1.4 Trace Gas and meteorological parameter measurements

158   The concentrations of trace gases (CO, O$_3$, SO$_2$, and NOx) were measured using gas monitors

159 (T400U, T100U, and T200U; Teledyne API Inc., USA). The meteorological elements, including

160 temperature, relative humility, wind speed, and wind direction, were measured by the combined

161 automatic weather station onboard the vessels.



**2.2 Data analysis**
**2.2.1 CCN activation**
The size-resolved number concentration of total praticle and cloud condensation nuclei were
obtained from the SMPS and CCNc thourgh the SMCA method. The activation diameter was determined
by fitting the activation ratio (AR, $N_{CCN}/N_{CN}$) and dry diameter at each supersaturation through the
following equation:
$$\frac{N_{CCN}}{N_{CN}} = \frac{B}{1+\left(\frac{D_P}{D_{50}}\right)^C},$$
    (1)

where $D_P$ represents dry particle diameter (nm); B, C, and $D_{50}$ are the three fitting parameters,
representing the asymptote, the slope, and the inflection point of the sigmoid, respectively (Moore et
al., 2010). The $D_{50}$ parameter, also known as the critical diameter, corresponds to the particle size at
which 50% of the particles are activated at a specific SS.
The hygroscopicity parameter (κ) which represents CCN activity according to κ-Köhler equation is
calculated as follows (Petters and Kreidenweis, 2007):
$$\kappa = \frac{4A^3}{27D_{50}^3(lnS_c)^2}, \; A = \frac{4\sigma_{s/a}M_w}{RT\rho_w}$$
    (2)

where $\rho_w$ is the density of pure water (about 997.04 kg m$^{-3}$ at 298.15 K), $M_W$ is the molecular weight of
water (0.018 kg mol$^{-1}$), $\sigma_{s/a}$ corresponds to the surface tension of the solution-air interface and is assumed
to be equal to the surface tension of pure water ($\sigma_{s/a}$=0.0728 N m$^{-1}$ at 298.15 K), R is the universal gas
constant (8.314 J mol$^{-1}$ K$^{-1}$), T denotes thermodynamic temperature in kelvin (298.15 K), and $D_{50}$ is the
critical diameter (in m).
**2.2.2 CCN concentration and activation ratio calculation**
The CCN concentration ($N_{CCN}$) can be predicted based on particle number size distribution (PNSD)
and $D_{50}$ at a specific SS. It can be calculated by the following equation (Cai et al., 2018):
$$N_{CCN}(SS) = \int_{D_{50}(SS)}^{\infty} N_{CN}(D_P)dD_p$$
    (3)

where $N_{CCN}$ (SS) is CCN concentration at a specific SS, $D_{50}$(SS) is the activation diameter at a specific
SS from the SMCA method and $N_{CN}(D_P)$ is the particle number concentration under specific diameter
from SMPS measurement.
The AR can be calculated by:



$$AR = \frac{\int_{D_{50}(SS)}^{\infty} N_{CN}(D_P) dD_p}{\int_0^{\infty} N_{CN}(D_P) dD_p} \qquad (4)$$
To investigate the effect of PNSD and $D_{50}$ on $N_{CCN}$ and AR, we defined delta $N_{CCN}$ ($\Delta N_{CCN}$) and
delta AR ($\Delta$AR). They are calculated by following equations:
$$\Delta N_{CCN} = \frac{N_{CCN,sim} - N_{CCN,actual}}{N_{CCN,actual}} \qquad (5)$$
$$\Delta AR = \frac{N_{CCN,actual} - N_{CCN,sim}}{N_{CN,actual}} \qquad (6)$$
The subscript "actual" represents the actual measured value, while the subscript "sim" represents the
simulated value.
**2.2.3 Primary and secondary organic carbon concentration calculation**
The concentrations of primary organic carbon (POC) and secondary organic carbon (SOC) were
calculated according to following equation:
$$POC = (OC/EC)_{pri} \times EC \qquad (7)$$
$$SOC = OC_{total} - (OC/EC)_{pri} \times EC \qquad (8)$$
where $(OC/EC)_{pri}$ is the OC/EC ratio in freshly emitted combustion aerosols, and $OC_{total}$ and EC are
available from ambient measurements. The $(OC/EC)_{pri}$ was obtained from the minimum R squared (MRS)
method (Wu and Yu, 2016). The MRS approach provides more accurate estimation of SOC than
$(OC/EC)_{min}$ or $(OC/EC)_{10\%}$ approach and the results obtained from the MRS method are sensitive to the
magnitude of measurement uncertainty, but the bias does not exceed 23% if the uncertainty is within 20%
(Wu and Yu, 2016). In this study, the $(OC/EC)_{pri}$ values were 3.65 and 0.25 in summer when affected by
terrestrial air masses and terrestrial-marine mixed air masses, while they were 2.82 and 0.82 in winter
when affected by terrestrial air masses and terrestrial-marine mixed air masses, respectively. The
categorization method based on the influence of different air masses will be introduced in the next section.
**2.2.4 Backward trajectory simulation**
Backward trajectory calculations were performed using MeteoInfo, an open-source software (Wang,
2014) to determine potential source origins. Weekly GDAS1 (Global Data Assimilation System at a
resolution of 1°) files were downloaded from the NOAA Air Resource Laboratory (ARL) website
(https://www.ready.noaa.gov/gdas1.php). The calculation of backward trajectories is performed every 12



hours based on the location of the ship, generating 48-hour backward trajectories at 50m, 150m, 500m,
and 1000m heights.
**3. Results and discussion**
**3.1 Overview**
Figure 2 shows the timeseries of PNSD (a1 and a2), $PM_1$ mass concentrations and fractions (b1 and
b2, c1 and c2), number concentrations of CCN (d1 and d2), and hygroscopicity κ-values (e1 and e2)
during two campaigns in summer and winter. Based on the backward trajectories and source origins, both
campaigns were classified into two periods: the period influenced by terrestrial-marine mixed air masses
(31%) and the period influenced by terrestrial air masses only (69%) (Fig. S3). During the summer,
terrestrial air masses primarily traverse through the Philippines region, whereas during the winter, they
predominantly originate from mainland China. As shown in Fig. 2, a higher total particle number
concentration during summer (11195 $cm^{-3}$) than during winter (6358 $cm^{-3}$) was obtained when the marine
atmosphere was mainly influenced by terrestrial air masses. This is in line with previous studies that a
higher particle number concentration of exceeding 15000 $cm^{-3}$ was observed in Manila (the largest port
city in the Philippines) than that of approximately 10000 $cm^{-3}$ reported in Guangzhou and Hong Kong
(Liu et al., 2008; Cai et al., 2017). Similar particle concentrations (2114 $cm^{-3}$ and 1840 $cm^{-3}$) in winter
and summer were shown when the marine atmosphere was predominantly influenced by mixed air
masses. Interestingly, they are much lower than those (~3400 $cm^{-3}$) observed in the northern South China
Sea (Cai et al., 2020), but higher than those observed in the remote SCS (975 $cm^{-3}$) when similar mixed
influences of air masses (Atwood et al., 2017).
The average mass concentration of NR-$PM_1$ was 3.60 μg $m^{-3}$ in summer and significantly increased
to 18.11 μg $m^{-3}$ in winter. However, this summer concentration was much lower than the concentrations
(9.11 μg $m^{-3}$ and 10.65 μg $m^{-3}$) measured in similar summer periods of 2018 and 2019 in the northern
SCS (Liang et al., 2021; Sun et al., 2023). As aforementioned, the particle number concentrations were
higher in summer than in winter when the atmosphere was primarily influenced by terrestrial air masses.
However, the particles in summer were predominantly in the Aitken mode, resulting in a lower
contribution to the mass loading than those in winter which were mainly in the accumulation mode (Fig.
3). Additionally, higher NR-$PM_1$ concentrations were found in both summer and winter under the



influence of terrestrial air masses than mixed ones, with decreases of approximately 0.6 µg m$^{-3}$ (from
3.84 to 3.20 µg m$^{-3}$) and 17 µg m$^{-3}$ (from 23.52 to 6.32 µg m$^{-3}$) in summer and winter, respectively. The
dramatic decrease for the latter case indicated a more pronounced impact of inland anthropogenic
emissions on the aerosol mass concentration in the SCS region during winter.

As shown in Fig. 2 (d1 and d2), the $N_{CCN}$ values at three supersaturation levels (0.2%, 0.4%, and

0.7%) in winter (1660, 2356, and 3053 cm$^{-3}$) were lower than those in summer (2899, 5450, and 5770
cm$^{-3}$), respectively. Specifically, the $N_{CCN}$ (5450 cm$^{-3}$) at 0.4% SS in this study during summer is much
higher than that (1544 cm$^{-3}$) measured at a similar SS (0.34%) in summer 2018 in the similar northern
SCS region (Cai et al., 2020). This is primarily due to most of the particles being in the Aitken mode
particles during summer (Figs. 3a and c). For comparison, the average total particle number concentration
in summer is twice that in winter, however, no such a significant difference in CCN concentration was
seen.

The median values of κ were approximately 0.5 in summer and 0.3 in winter (Fig. 2e), respectively.

During summer, the median κ value was slightly higher than that (~ 0.4) measured in 2018 in the similar
northern SCS (Cai et al., 2020), but lower than that (0.65) in the southern SCS (Atwood et al., 2017).
Comparatively, under the influence of terrestrial air masses, the κ values ranged from 0.41 to 0.50 (Fig.
3a), close to the above value of 0.4 in 2018 (Cai et al., 2020). Furthermore, under the influence of mixed
air masses, the κ values ranged from 0.49 to 0.61 (Fig. 3c), similar to those observed in the southern SCS
(0.54) and the Western North Pacific (0.49–0.64), but lower than those in the southern SCS (0.65) and
the Western North Pacific (0.48–0.84) under the influence of marine air masses (Atwood et al., 2017;
Kawana et al., 2022). During winter, under the influence of terrestrial air masses, the κ values ranged
from 0.17 to 0.38 (Fig. 3b), very close to those (0.14–0.37) measured in Guangzhou (Fig. 3d) (Cai et al.,
2018). This specific winter scenario indicates that aerosols in the northern SCS are significantly
influenced by the air masses transported from the mainland China during winter season. Compared to
terrestrial air masses, the κ values were respectively higher at the set SS under the influence of mixed air
masses (Fig. 3d). The above differences on hygroscopicity are likely attributed to variations in aerosol
chemical composition during different seasons and under the influence of different air masses. In both
winter and summer, the PNSD exhibits a unimodal distribution under the influence of terrestrial air
masses, while it shows a bimodal distribution under the influence of mixed air masses (Fig. 3). The
bimodal distribution may result from a mixture of particles from marine primary emissions and from



transported anthropogenic emissions across the region, while the unimodal distribution was primarily
composed of particles from inland-transported anthropogenic emissions (Frossard et al., 2014; Atwood
et al., 2017).

**3.2 Impact of chemical composition on hygroscopicity**

Chemical composition is a crucial factor influencing aerosol hygroscopicity due to distinct
hygroscopic properties exhibited by each chemical component (Petters and Kreidenweis, 2007). We
observed two significant differences in aerosol hygroscopicity between summer and winter (Fig. 3).
Firstly, aerosol hygroscopicity was considerably higher in summer than that in winter. Secondly, aerosol
hygroscopicity under terrestrial-influencing air masses was lower than that under mixed-influencing air
masses. Additionally, hygroscopicity decreased with increasing particle size in summer, whereas
opposite effect was observed in winter. In this section, we primarily discuss the differences in aerosol
hygroscopicity between summer and winter, as well as the variations observed when influenced by
different air masses, from the chemical composition perspective.

**3.2.1 Impact of inorganic components**

In summer, the highest proportion of NR-PM$_1$ components measured by ACSM is sulfate (40.3%),
followed by organic (37.3%), ammonium (14.0%), nitrate (6.2%), and chloride (2.3%) (Fig. 4). However,
in winter, organic (39.2%) has the highest proportion, followed by nitrate (22.1%), sulfate (19.5%),
ammonium (17.7%), and chloride (1.5%) (Fig. 4). Compared to summer, the significant increase of
nitrate proportion and decrease of sulfate in winter were mainly attributed to the difference in air masses.
Winter terrestrial air masses originate from mainland China and primarily pass through the PRD region
to reach the South China Sea, while the summer ones primarily originate from the Philippines region.
Stronger transportation and industrial emissions in the PRD than in the Philippines lead to higher NOx
concentrations and hence higher nitrate concentrations (Wang et al., 2019). A high proportion of nitrate
sulfate in the PRD during winter was previously reported in NR-PM$_1$ (Yang et al., 2022), whereas sulfate
was found the major component in the Philippines (Tseng et al., 2019).
Previous studies showed that nitrate has slightly higher hygroscopicity than sulfate (Gysel et al.,
2007; Topping et al., 2005). For example, the κ values for ammonium sulfate ((NH$_4$)$_2$SO$_4$), ammonium
bisulfate (NH$_4$HSO$_4$), and ammonium nitrate (NH$_4$NO$_3$) are 0.48, 0.56, and 0.58, respectively (Huang et



al., 2022). In some marine environments, aerosols exhibit a high acidity with low ammonium
concentrations, where sulfate and nitrate primarily exist in the form of sulfuric acid ($H_2SO_4$, κ: 0.7) and
nitric acid ($HNO_3$, κ: 0.85), respectively, thereby enhancing aerosol hygroscopicity (Chang et al., 2011;
Siegel et al., 2022). Here, we use the ratio of $NH_4^+$ (measured) to $NH_4^+$ (predicted) to roughly determine
an aerosol being "acidic" or "alkaline" (Guo et al., 2015): more acidic if this ratio is smaller one because
the imbalance indicates the possible presence of acidic compounds such as $H_2SO_4$, $HNO_3$, or HCl in the
aerosol (Zhang et al., 2021). The median values of this ratio in summer and winter are 0.82 and 1.11,
respectively, exceeding the threshold of 0.75 (Fig. S5), a rough estimate for equal molecular numbers of
$(NH_4)_2SO_4$ and $NH_4HSO_4$. This indicates that sulfate and nitrate are primarily present in the form of
$(NH_4)_2SO_4$ and $NH_4NO_3$ both in summer and winter in the SCS. Since the hygroscopicity values of
$(NH_4)_2SO_4$ and $NH_4NO_3$ are similar, changes in their proportions may not be the primary driver of the
observed aerosol hygroscopicity variations between winter and summer. Interestingly, we found that the
mass concentrations of BC and EC were higher in winter (1.79 μg m$^{-3}$ and 1.16 μg m$^{-3}$) than those in
summer (0.61 μg m$^{-3}$ and 0.64 μg m$^{-3}$), which might be one of the reasons for the lower aerosol
hygroscopicity in winter because of the hydrophobicity nature of BC (κ=0) (Weingartner et al., 1997;
Siegel et al., 2022). Another reason may be attributed to the hygroscopicity variations of organic
components ($κ_{org}$), which will be discussed in the following section.

As introduced at the beginning of the section, the hygroscopicity decreased with increasing particle

size during summer, in contrast to the increasing trend during winter, which may be attributed to the
oxidation of dimethyl sulfide (DMS). DMS oxidation produces methanesulfonic acid (MSA) which is
further oxidized to form non-sea salt (NSS) sulfate in smaller particles (Raes et al., 2000). The ratio of
sulfate to MSA (SA/MSA) can be used to assess the impact of DMS oxidation and anthropogenic
emissions on sulfate (Savoie et al., 2002; Zhang et al., 2007; Cai et al., 2020). A lower ratio indicates a
more significant influence of DMS oxidation on sulfate, while a higher ratio suggests a greater influence
of anthropogenic emissions. Previous studies have found that SA/MSA is approximately 15–625 in the
northern South China Sea region, significantly higher than that (18–20) in the open ocean (Zhang et al.,
2007; Savoie et al., 2002). Cai et al. (2020) reported this ratio in a range of 100–10000 using Modern-
Era Retrospective analysis for Research and Applications, Version 2 (MERRA-2) reanalysis data in a
similar northern SCS region. In this study, since we could not accurately distinguish MSA from other



organic components using Tof-ACSM due to its low mass resolution, we employed the same method to
analyze the SA/MSA distribution in the South China Sea region during the observation period (Fig. S6).
Our results showed that this ratio ranged from approximately 100 to 3000 in summer, whereas it exceeded
10000 in winter, indicating a greater impact of anthropogenic emissions than DMS oxidation on the NSS
sulfate in the SCS, with a more pronounced influence in winter than in summer. Higher DMS
concentrations in summer than in winter are likely due to higher sea surface temperatures which promote
the production rate of DMS by phytoplankton (Bates et al.,1987). Therefore, it is likely that there is a
higher amount of NSS sulfate produced through DMS oxidation during summer, contributing to the
sulfate production and hence increasing aerosol hygroscopicity in smaller particles during summer.
**3.2.2 Impact of organic components**
The mass concentrations of organics exhibit significant differences between winter (7.15 μg m$^{-3}$)
and summer (1.34 μg m$^{-3}$), while the total fractions in NR-PM$_1$ remain relatively consistent (40% and
37% in winter and summer, respectively). Despite their similar proportions in winter and summer, the
differences in the organic components may likely play an important role in the variations of
hygroscopicity. Previous studies showed that the measured hygroscopic parameter was linearly
correlated with the organic mass fraction (f$_{org}$) (Bhattu et al., 2016; Huang et al., 2022; Dusek et al., 2010).
Here, we employed linear regression to relate κ obtained at 0.2% SS (κ$_{ss:0.2\%}$) to f$_{org}$ measured by ToF-
ACSM (Fig. 5). Two reasons for choosing κ$_{ss:0.2\%}$: (1) It represents hygroscopicity of larger particles
(~100 nm) than that at higher SSs. (2) The κ values at 0.1% SS (κ$_{ss:0.1\%}$) is only available in winter. In
addition, the measured mass concentrations focused on larger particles which were represented by those
of κ$_{ss:0.2\%}$. We consider κ as hygroscopicity of organics (κ$_{org}$) when f$_{org}$ approaches 1 and as that of
inorganics (κ$_{inorg}$) when f$_{org}$ is nearly 0.
During summer and winter under the influence of terrestrial air masses, the κ$_{org}$ value was
approximately 0.09, while under the influence of mixed air masses, it was approximately 0.21 and 0.20,
respectively. This result suggests that no significant differences of κ$_{org}$ were found between summer and
winter under the same influence of air masses. However, pronounced differences of κ$_{org}$ were indeed
observed under the influence of different air masses which may be associated with different types of
organic compounds responsible for hygroscopicity. This was further supported by variations of the POC
and SOC fractions. In general, POC is associated with primary organic aerosol (POA), primarily



originating from combustion sources, such as fossil fuel, biomass burning, and biogenic emissions, while
SOC is associated with secondary organic aerosol (SOA), primarily originating from the oxidation of
POA (Wu and Yu, 2016). In addition, SOC exhibits higher hygroscopicity than POC (Jayachandran et
al., 2022; Safai et al., 2014). Previous studies have indicated that highly aged organic compounds exhibit
higher hygroscopicity than the less oxidized counterparts (Lambe et al., 2011; Jimenez et al., 2009).
Previous studies showed that the $\kappa_{org}$ values are in a range of 0.02–0.17 when the O/C ratio varies from
0.2 to 0.6 (Jimenez et al., 2009) and $\kappa_{POA}$ ($\kappa$ corresponds to POA, similar to $\kappa_{org}$) and $\kappa_{SOA}$ are respectively
in the ranges of 0–0.1 and 0.1–0.3 due to different types of organic aerosol (Liu and Wang, 2010; Kuang
et al., 2020). A higher proportion of POC was observed during the terrestrial periods, whilst a significant
increase in the fraction of SOC was shown under the influence of mixed air masses (Figs. 4 h-m). As a
result, the increasing oxidation degree of SOA and the decreasing proportion of POA lead to higher $\kappa_{org}$
under mixed air masses than those under continental air masses. This is also consistent with a recent
study in the SCS (Sun et al., 2023), revealing that a combined effect of photo-oxidation and liquid-phase
reactions play an essential role in the BBOA aging processes.

**3.3 Differences in the CCN activation ratio during different seasons**

The CCN activation ratio (AR), which quantifies the number fraction of aerosol particles capable
of acting as CCN at a specific supersaturation level, is an important parameter for characterizing the
CCN activity (Dusek et al., 2006). In addition, AR is also a useful parameter in the CCN prediction
(Pruppacher, 2010; Deng et al., 2013). Figure 6 illustrates the variations in AR and $N_{CCN}$ under the
influence of different air masses during different seasons. The median AR values were higher in summer
(0.39, 0.67, and 0.85 at 0.2%, 0.4%, and 0.7% SS, respectively) than in winter (0.21, 0.36, 0.49, and 0.64
at 0.1%, 0.2%, 0.4%, and 0.7% SS, respectively). In summer under terrestrial air masses, the ARs (0.34–
0.81 at 0.2%–0.7% SS) in this study were lower than those (0.49–0.85 at 0.18%–0.59% SS) in a previous
study in the same region (Cai et al., 2020), but higher than those (0.18–0.48 at 0.11%–0.60% SS) in the
Western North Pacific (Kawana et al., 2022). However, under mixed air masses, the AR values were
higher than those (0.31–0.71 at 0.18%–0.59% SS) reported in Cai et al. (2020) and close to those (0.42–
0.85 at 0.11%–0.60% SS) in the Western North Pacific (Kawana et al., 2022). In winter under terrestrial
air masses, the AR values were in a range of 0.14 to 0.59 at 0.1%–0.7% SS, consistent with those





measured in the Guangzhou (0.26–0.64 at 0.1%–0.7% SS) and Hong Kong (0.16–0.65 at 0.15%–0.7%
SS) (Cai et al., 2018; Meng et al., 2014).

The differences of AR between different seasons under different air masses could be attributed to

different PNSD and hygroscopicity. When the PNSD concentrates particles in the size range beyond the
$D_{50}$, more particles can be activated as CCN, leading to higher ARs. Meanwhile, more hygroscopic
particles intend to have smaller $D_{50}$ values with which more particles can be activated. Previous studies
showed that the influence of the above two factors may vary under different environments. The PNSD
played a more important role in most cases in continental area (Dusek et al., 2006; Tao et al.,2021), while
hygroscopicity can also have significant effects in some environments, such as boreal forest and coastal
area (Roldin et al., 2019, Bougiatioti et al., 2016). Here, we investigate the relative importance of aerosol
PNSD and chemical composition (hence hygroscopicity) in the SCS during different seasons under
different air masses. We define ΔAR as the difference between the actual and estimated AR using Eq.
(7), following a procedure illustrated in Fig. S8. AR was calculated using Eq. (4) based on the average
$D_{50}$ and the PNSD obtained during different seasons and periods (Fig. S8). Figure 7 shows that the ΔAR
values at 0.2%–0.7% SS between summer and winter due to $D_{50}$ were from -22% to 29%, while they
were from -10% to 12% due to PNSD, indicating more significant influence of $D_{50}$ than PNSD on $N_{CCN}$
in the South China. In fact, the peak diameters of PNSD during both winter and summer are around 70
nm, and their variations are relatively small compared to the changes in $D_{50}$ (Fig. S9).

We further investigate the influence of PNSD and hygroscopicity on AR under different air masses

in the same season. In summer, the influence of PNSD on the AR can reach 6.0% to 9.8% between
different periods (Figs. 7 b1 and c1), with the most significant impact at 0.2% SS ($D_{50}$ > 90 nm). This
can be attributed to the fact that large variations in accumulation mode particles significantly impact AR
with peaking at a larger size for terrestrial air masses (151 nm) than mixed air masses (140 nm) (Figs. S9
a1-a3). In comparison, the impact of hygroscopicity (based on $D_{50}$) on AR is relatively small, for example,
the influence is estimated to be around 1.3%–3.4% at 0.2% and 0.7% SS, and ~6% at 0.4% SS. In winter,
the impact of hygroscopicity on AR at all SS levels (2.6%-5.1%) was more significant than that of PNSD
(1.2%–3.2%) (Fig. 7). We attribute this difference to two possible reasons: (1) Particles tend to be more
concentrated in larger sizes (Fig. S9) and the changes in PNSD under different air masses have a
relatively minor impact on AR. (2) the $D_{50}$ values in winter are larger than those in summer and are closer





to the peak sizes of the PNSD (Figs. S9 b1-b3). The results suggest that PNSD was the most important

factor influencing $N_{CCN}$ in summer, while hygroscopicity became more important in winter.

**3.4 Influence of spatial distribution of particle properties on $N_{CCN}$**

This section focuses on how the spatial distribution of aerosol concentration and physicochemical

properties influences CCN concentrations under different air masses, particularly the variations occurring

with offshore distance. Typically, $N_{CCN}$ in a specific region is predicted in models with a single

parameterized PNSD and it can be used to simulate the hygroscopic growth process (Yu and Luo, 2009).

Here, we further investigated the impact of spatial variations in PNSD and hygroscopicity on the

prediction of $N_{CCN}$. In summer, terrestrial air masses primarily pass through the Philippines and the

impact varies with longitude, while during winter, terrestrial air masses are predominantly from mainland

China, especially the PRD region, and the impact varies with latitude (Fig. S3). Hence, we investigate

the influence in summer based on longitude and in winter based on latitude. Figure 8 illustrates the

variations of the $N_{CCN}$ and number concentration of different mode particles, κ at 0.4% SS ($κ_{ss:0.4\%}$), the

ratio of SOC to OC, mass concentration of BC, and NR-PM$_1$ component proportions with increasing

offshore distance.

In summer, the $N_{CCN}$ showed a decreasing trend with increasing offshore distance under the

terrestrial air masses, while no significant variation was observed under the mixed air masses.

Furthermore, the number concentration of Aitken mode particles significantly decreased with increasing

offshore distance (from 118°E to 113°E), from nearly $10^4$ to approximately $10^3$ (Fig. 8a2). Similarly, the

number concentration of accumulation mode particles exhibited a decreasing trend. In contrast, no

significant variation was observed for the number concentration of nucleation mode particles, indicating

that those particles were less affected by the terrestrial air masses. The $κ_{ss:0.4\%}$ values exhibit an increasing

trend with increasing offshore distance under the terrestrial air masses, while no obvious trend was shown

under the mixed air masses (Figs. 8 a3 and b3). The above observations can be attributed to three possible

reasons: (1) A higher degree of oxidation for the organic component enhances the hygroscopicity of the

particles, as evidenced by the increasing trend of the SOC/OC ratio with increasing offshore distance

(0.4 to 0.7 from 118°E to 113°E, Fig. 8a4). (2) A significant decrease in the BC concentration (0.5 to 0.3

μg m$^{-3}$ from 118°E to 113°E, Fig. 8a5) may contribute to an increase of the aerosol hygroscopicity. (3)

An increase of sulfate fraction with increasing offshore distance may lead to a decrease of organic



compounds (Fig. 8a6). The increase of hygroscopicity with increasing offshore distance lowered the $D_{50}$
values and led to the increase of $N_{CCN}$, while the decrease of $N_{CCN}$ was primarily attributed to the change
of the particle concentrations in Aitken mode and accumulation mode.

In winter, the increasing trend with increasing offshore distance is similar to that in summer under

the terrestrial air masses, while no significant $N_{CCN}$ variation was observed under the mixed air masses
(Figs. 8c1-c2 and d1-d2). Meanwhile, the aerosol hygroscopicity exhibits a slight increasing trend with
offshore distance (a $\kappa$ value of 0.19 to 0.21 from 22°N to 19°N). The increase of aerosol hygroscopicity
with increasing offshore distance can be explained by two possible reasons: (1) The nitrate mass fraction
decreased significantly and organics became the predominant component with increasing offshore
distance. Hence, the increased oxidation degree of the organic component was an important factor for
the enhancement of aerosol hygroscopicity which is supported by the increase of the SOC/OC ratio (Fig.
8c4). (2) The decrease of the BC concentrations with increasing offshore distance was more pronounced
during winter (2.1 to 0.6 $\mu$g m$^{-3}$) than during summer (0.5 to 0.3 $\mu$g m$^{-3}$), indicating that the decrease of
the BC mass fraction is also a significant contributor to the overall increase of aerosol hygroscopicity.
Interestingly, the slight decrease of the average $\kappa$ value further offshore (19°N to 20°N) was caused by
the impact of nearby ship emissions which was further supported by the decrease of SOC/OC ratio.

The impact of spatial distribution of $D_{50}$ and PNSD on $N_{CCN}$ was further investigated through

calculating $N_{CCN}$ using $D_{50}$ or PNSD (Eq. (3)) according to the scheme shown in Fig. S10. For example,
to investigate the impact of PNSD on $N_{CCN}$, the PNSD which are closest and furthest from the shore and
the observed $D_{50}$ were used to calculate $N_{CCN}$. Here, we define $\Delta N_{CCN}$ as the ratio of the difference
between calculated $N_{CCN}$ and observed $N_{CCN}$, divided by the observed $N_{CCN}$ (Eq. (7)).  However, under
the mixed air masses, as mentioned above, since no obvious variations of PNSD and aerosol
hygroscopicity with offshore distance were observed (Figs. 8 b2-b3 and d2-d3), the impact of spatial
distribution of $D_{50}$ and PNSD on $N_{CCN}$ was less significant (Fig. S11). Hence, below we only discuss the
scenarios under the terrestrial air masses.

Figure 9 shows $\Delta N_{CCN}$ at different offshore distances under the terrestrial air masses. In summer, a

monotonic trend of $\Delta N_{CCN}$ was observed with a value of up to approximately 70% at 0.2 SS under the
effect of hygroscopicity. In winter, the impact of hygroscopicity was less pronounced than that in summer,
reaching only up to around 10% (Figs. 9 c1 and c2). This indicates that a single hygroscopicity parameter



can represent the overall hygroscopicity in SCS under the influence of terrestrial air masses in winter,
whereas during summer, this approach may not be sufficient. The $\Delta N_{CCN}$ nearshore caused by the impact
of PNSD can exceed 300% and can reach up to 80% furthest offshore during summer, indicating that the
spatial distribution of PNSD has a greater impact on $N_{CCN}$ than hygroscopicity. Similarly, although the
impact was less significant during winter, the $\Delta N_{CCN}$ nearshore can still reach 80% (Fig. 9 d1 and d2).
Hence, in summary, using a single PNSD for the representation of PNSD in the region of SCS can lead
to significant uncertainties in the prediction of $N_{CCN}$ under the terrestrial air masses during both summer
and winter.
**4. Conclusion**
In this study, we investigated the seasonal variations of CCN activity in the SCS and explored the
impact of anthropogenic emissions, based on shipborne observations conducted during the summer (May
5–June 9) and winter (December 19–29) of 2021. CCN activity, chemical composition, and particle
number size distribution (PNSD) over the SCS were measured using several onboard instruments
including a ToF-ACSM, a CCNc, an SMPS, an OC/EC analyzer, an AE33, and several monitors for trace
gases (i.e., $SO_2$, NOx, CO, and $O_3$). Our results show that the particle number concentration ($N_{CN}$) and
CCN number concentration ($N_{CCN}$) during summer were higher than those during winter, while the NR-
$PM_1$ mass concentration was lower in summer, which can be attributed to the predominance of the Aitken
mode particles in summer than a significant higher concentration of the accumulation mode particles
during winter. Additionally, the aerosol hygroscopicity was found to be higher in summer than in winter,
likely due to the enhanced terrestrial air masses and the increased proportions of black carbon and
decreased sulfate concentrations from DMS oxidation.
Based on backward trajectories, both campaigns could be divided into two periods: period affected
by terrestrial air masses and period affected by both terrestrial and marine (mixed) air masses. In summer,
the terrestrial air masses originate primarily from the Philippines, whereas in winter, they predominantly
originate from the PRD region of China. The hygroscopicity under terrestrial air masses was lower than
that under mixed air masses during both seasons, and it was similar to that observed in the PRD region
during winter. The PNSD distribution exhibited a unimodal pattern under the terrestrial air masses,
whereas a bimodal pattern was observed under the mixed air masses. During winter, the $N_{CCN}$ values



were higher under terrestrial air masses in contrast to higher AR values under mixed air masses. During
summer, the AR ratio was primarily influenced by PNSD, whereas during winter, it was more strongly
influenced by aerosol hygroscopicity than by PNSD.

Our study demonstrated significant variations in the aerosol concentration and physicochemical

properties with increasing offshore distances under terrestrial air masses during both summer and winter.
The decreasing trends observed in various gas concentrations ($NO_X$ and CO) with increasing offshore
distances suggest a diminishing influence of anthropogenic emissions. The aerosol hygroscopicity
increased with increasing offshore distances, primarily due to the decrease of the organic fraction, the
oxidation degree of the organic component, the decreased proportions of black carbon, and the increased
sulfate ratio. We found that the predicted CCN concentrations nearshore based on a single PNSD under
terrestrial air masses could lead to significant error (15%–360%). In contrast, using a representative
hygroscopicity parameter value nearshore can reduce the error in predicting $N_{CCN}$ (5%–10%). Hence, the
PNSD had a greater impact on $N_{CCN}$ prediction than hygroscopicity. Our study highlights the significant
differences of CCN activity during summer and winter in the SCS and significant influence of
anthropogenic emissions on the CCN activity. Future studies should include observations during spring
and autumn to explore the impact of mixing state on aerosol hygroscopicity for a more comprehensive
understanding of CCN activity in this region.

*Data availability.* Data from the measurements are available at
https://doi.org/10.6084/m9.figshare.25472545 (Ou et al., 2024).

*Supplement.* The supplement related to this article is available online at xxx.

*Author contributions.* **HO, MC, and JZ** designed the research. **YZ, XN, BL, and CS** performed the
measurements. **HO, MC, QS, and SM** analyzed the data. **SZ and HW** provided useful comment on the
paper. **HO, MC,** and **JZ** wrote the paper with contributions from all co-authors.

*Competing interests.* The authors declare that they have no conflict of interest.



*Financial support.* This work was supported by National Natural Science Foundation of China (NSFC)
(Grant No. 42305123 and 42175115) and Basic and Applied Basic Research Foundation of Guangdong
Province (Grant No. 2023A1515012240).

*Acknowledgements.* Additional support from the crew of the vessels "Tan Kah Kee" and "Sun Yat-sen
University" is greatly acknowledged.






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



FIGURE CAPTION
Figure 1. The cruies of two shipborne observation.
Figure 2. Timeseries of (a) particle number size distribution, (b) mass concentration of NR-PM1, and (c)
its fraction, (d) mass concentration of organic carbon and elemental carbon, (e) number concentration of
total particle and cloud condensation nuclei under the supersaturation of 0.1%, 0.2%, 0.4%, and 0.7%,
and (f) aerosol hygroscopicity. The number 1 means timeseries in summer and number 2 means it in
winter.
Figure 3. Particle number size distribution under effect of terrestrial air masses in summer (a) and winter
(b); Particle number size distribution under effect of mixed air masses in summer (c) and winter (d); The
red markers represent the activation diameters and hygroscopicity parameters corresponding to 0.1%,
0.2%, 0.4%, and 0.7% supersaturations in this study (without 0.1% in summer). The green markers
represent the hygroscopicity parameters reported in Atwood et al. (2017) for the southern South China
Sea during summer. The gray markers represent the hygroscopicity parameters documented in Cai et al.
(2018) for the Pearl River Delta region during winter.
Figure 4. The mass concentration of NR-PM1 , primary organic carbon, secondary carbon, elemental
carbon, and black carbon and their fraction under effect of different air masses in summer and winter.
Figure 5. Scatter plot of κ under the supersaturation of 0.2% and organic mass fraction with linear
regression.
Figure 6. Activation ratio at supersaturation of 0.1%, 0.2%, 0.4%, and 0.7% under effect of terrestrial
and mixed air masses in summer and winter. The box extends from the first quartile (Q1) to the third
quartile (Q3) of the data, with a line at the median. The box extends from Q1 to Q3 of the data, with a
line at the median. The whiskers extend from the box by 1.5 times of the inter-quartile range (IQR). Flier
points are those passing the end of the whiskers.
Figure 7. Differences in activation fraction calculated from different particle size distributions and
activation diameters.
Figure 8. Variations of κSS:0.4%, trace gases (NOx and CO), OC/EC, BC, SOC/OC, mass fraction of
NR-PM1 chemical composition, as well as number concentrations of nucleation mode, Aitken mode, and
accumulation mode with offshore distance under effect of terrestrial and mixed air masses in summer
and winter.





Figure 9. The difference between the calculated cloud condensation nuclei number concentration (NCCN)
using the D50 from the farthest and nearest offshore distances, along with the measured particle number
size distribution (PNSD), and the measured NCCN under the effect of terrestrial air masses in summer
(a1 and a2) and winter (c1 and c2); The difference between the calculated NCCN using the PNSD from
the farthest and nearest offshore distances, along with the measured D50, and the measured NCCN under
the effect of terrestrial air masses in summer (b1 and b2) and winter (d1 and d2). ΔNCCN refers to the
difference between calculated NCCN and observed NCCN, divided by the observed NCCN.




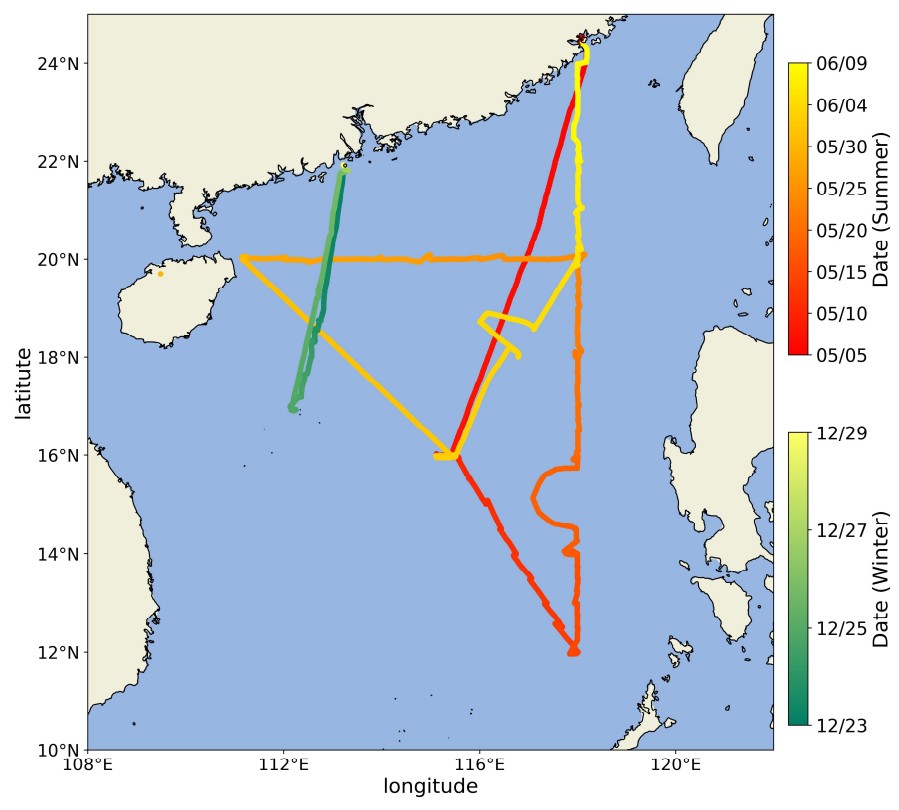

Fig. 1



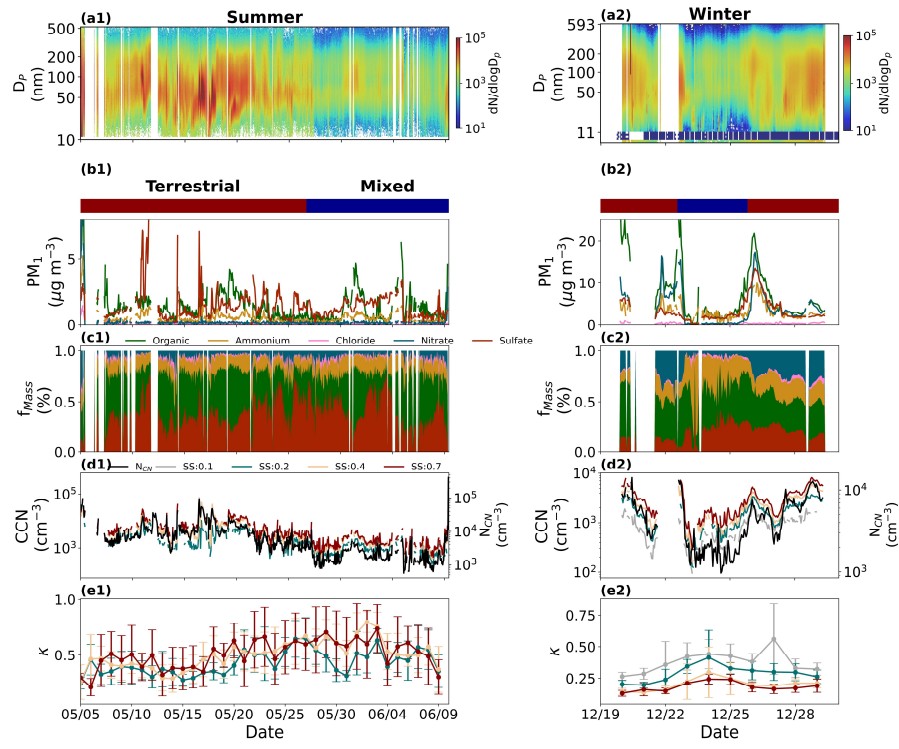


Fig. 2



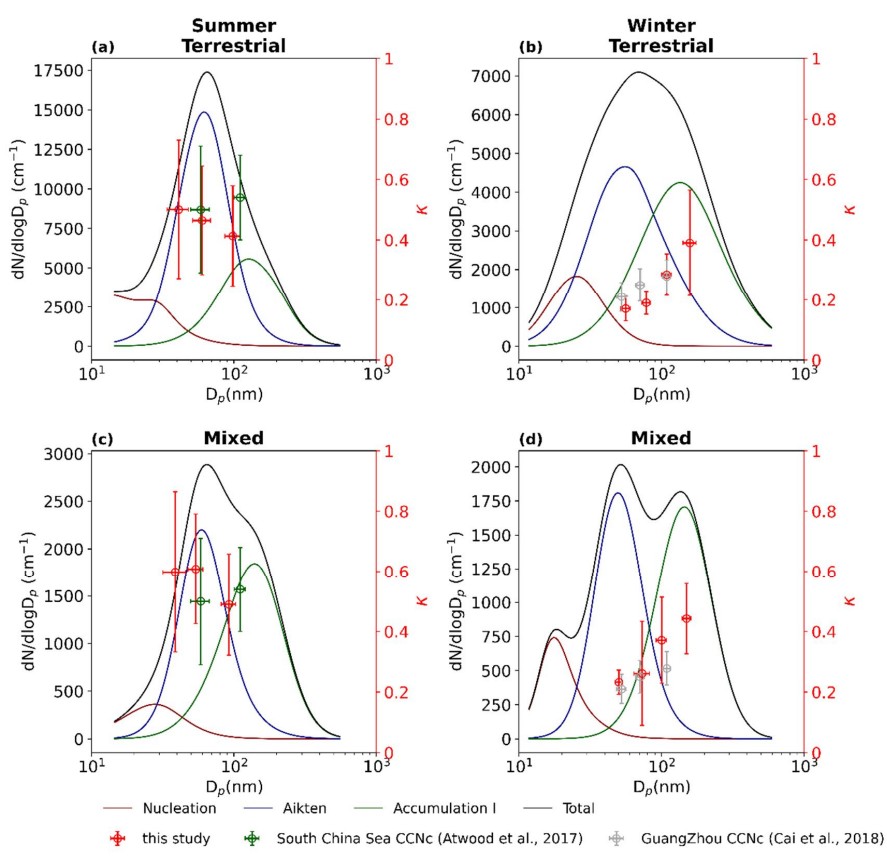


Fig. 3



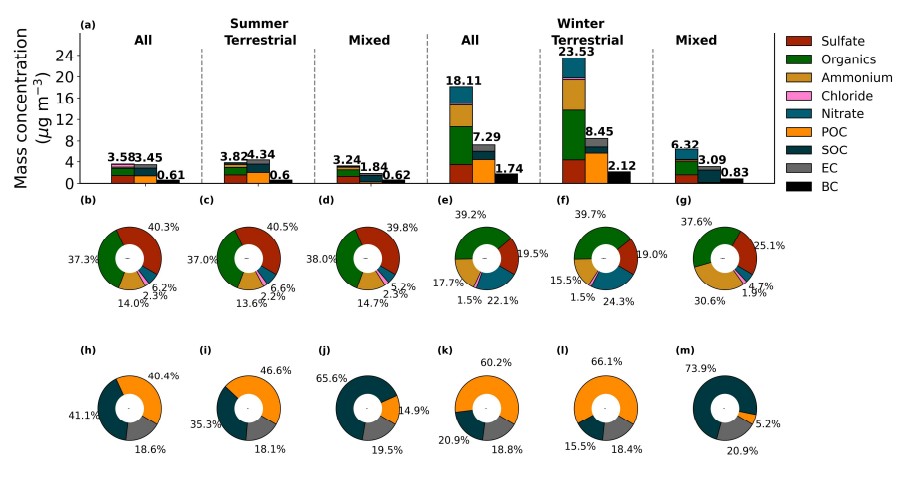



Fig. 4



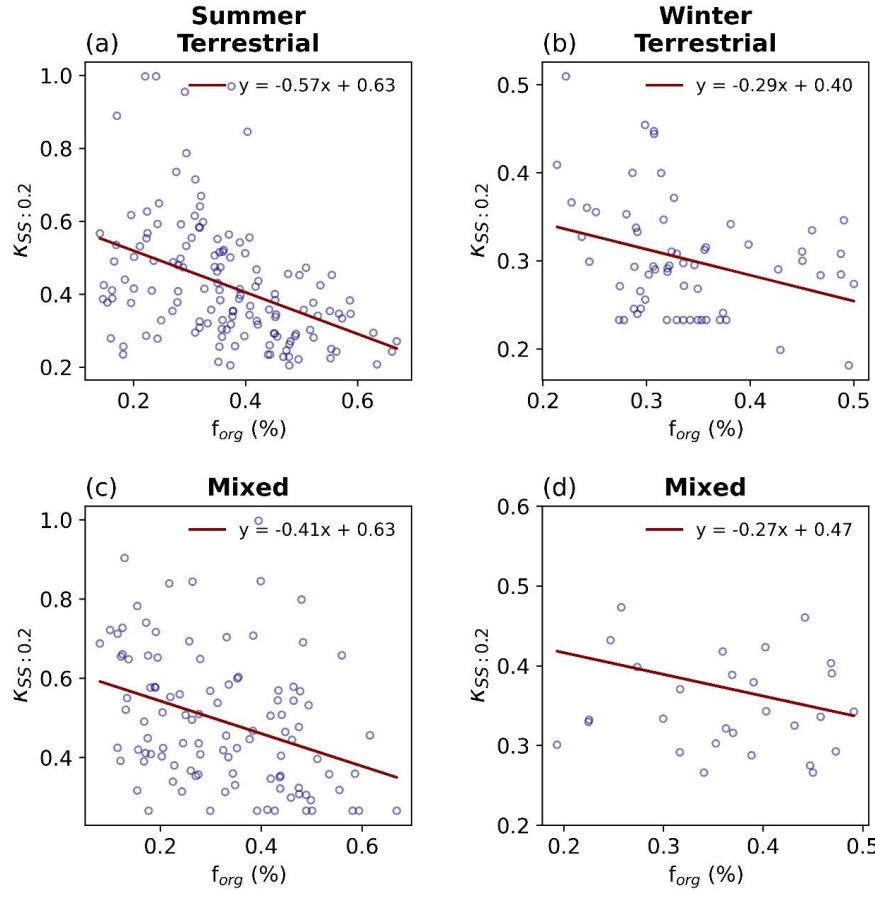


Fig.5




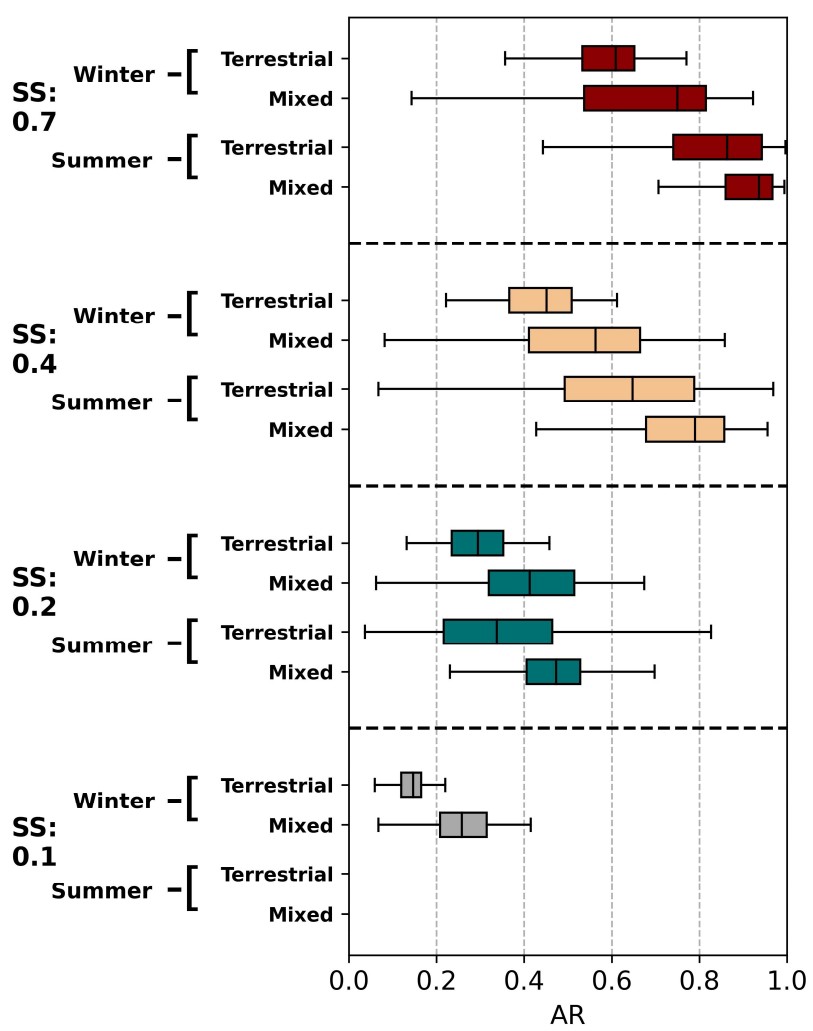

Fig. 6



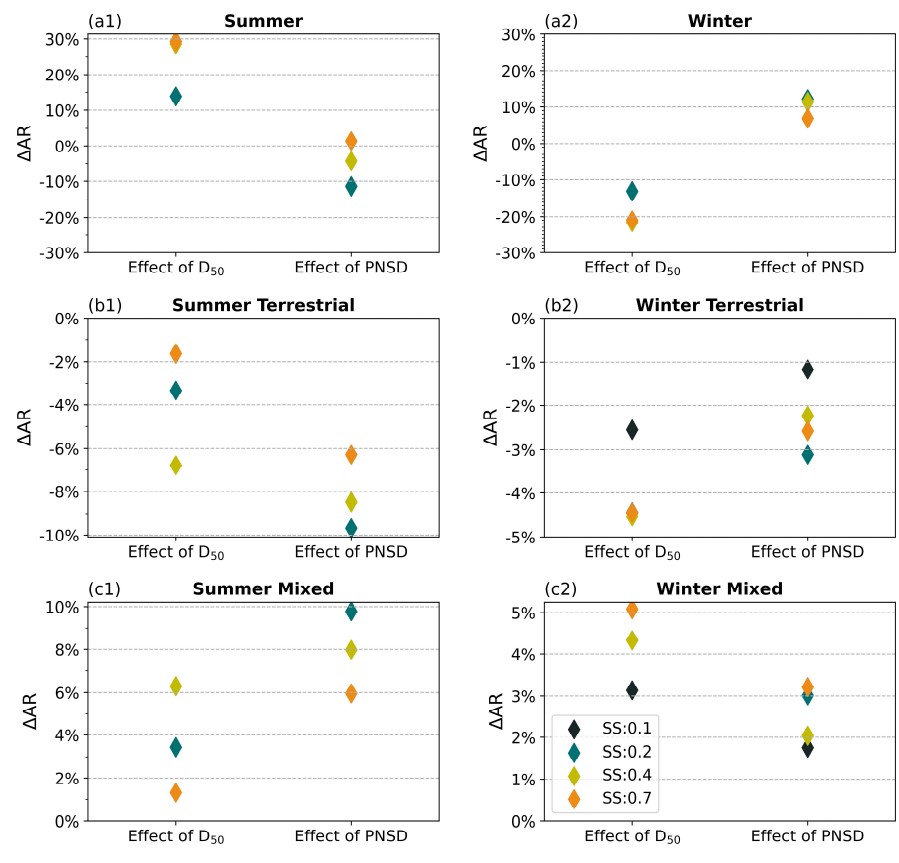


Fig. 7





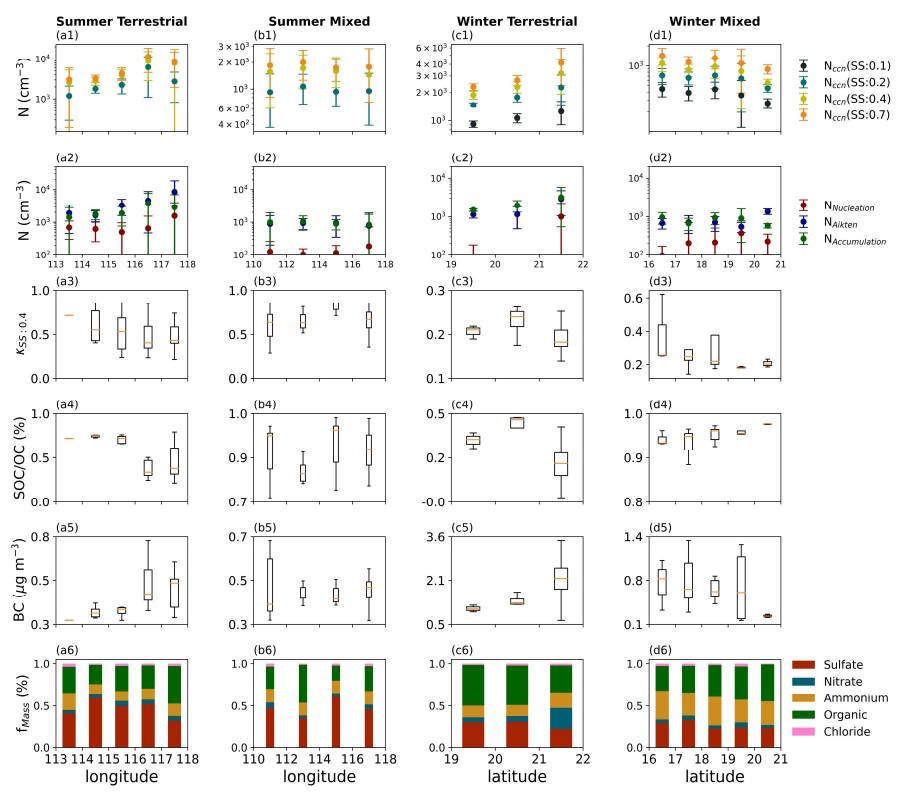


Fig. 8



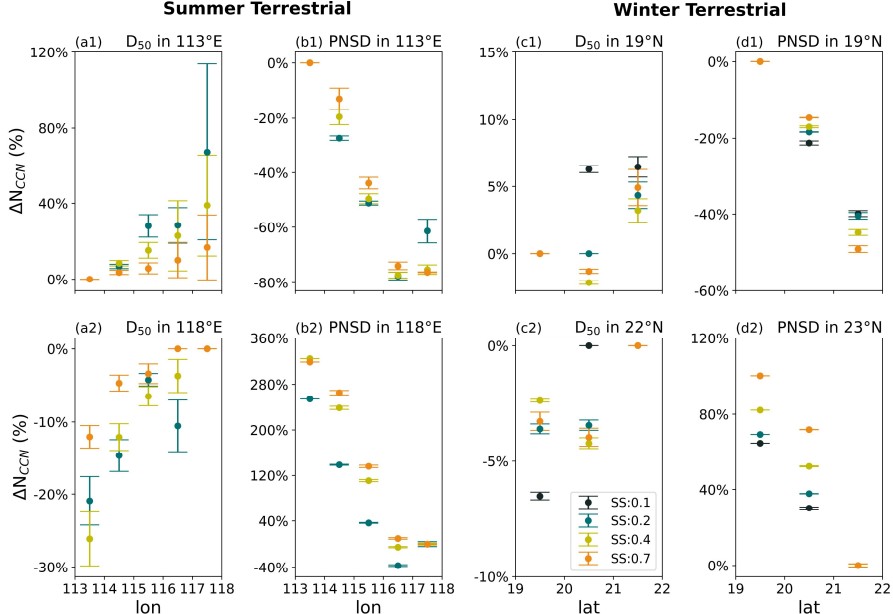

Fig. 9