# Peer review of "Measurement Report: Cloud condensation nuclei (CCN)"

_EGUsphere, 2024_

## Referee Comment (RC1)

Reviewer's Comment

Manuscript Number: **Egusphere-2024-956**

The manuscript presents a study on cloud condensation nuclei (CCN) activity in the South China Sea (SCS), examining seasonal variations and anthropogenic influences through two shipborne campaigns in the summer and winter of 2021. The researchers measured aerosol chemical composition, particle number size distribution (PNSD), and CCN, revealing significant seasonal differences in aerosol properties and the impact of anthropogenic emissions on CCN activity. Despite these insights, the manuscript appears hastily written, lacking novelty, and presents vague data and discussions that are difficult to interpret. Furthermore, the manuscript fails to clearly differentiate these campaigns from previously reported measurements in the SCS, reducing its contribution to the existing body of knowledge.

The manuscript is challenging to follow, primarily describing a lengthy dataset with confusing correlations and writing that diminish readability. The main scientific conclusions are unclear, and the text is overly lengthy, obscuring the scientific merits of the data analysis. It is recommended that the authors focus on a few key scientific findings in their revision. Based on the given title, the manuscript should focus solely on reporting the measurements from the cruise campaigns. However, it also delves into scientific analysis, giving the impression that the authors aim to publish their analytical findings under the guise of reporting measurements. This approach detracts from the clarity and purpose of the study, as the title suggests a primary emphasis on measurement data rather than comprehensive scientific interpretation. In its current form, the manuscript is not suitable for publication in ACP.

**Major comments**:

Regarding the main scientific findings presented in the manuscript, the discussion does not adequately support these claims. The authors assert that they represent the seasonal variation of aerosol-CCN activity; however, this is not convincingly demonstrated. The winter observations span only 10 days and, unlike the summer campaign, do not spatially cover the entire region. Furthermore, there is no discussion about the potential bias introduced by the limited sampling, which could significantly affect the derived results. Addressing this gap is crucial for validating the study's conclusions.

While the title effectively highlights the focus on seasonal variations in CCN activity in the South China Sea (SCS), the Introduction needs further development for full contextualization. It should provide detailed background on the SCS's unique meteorological and environmental characteristics, such as meteorology and anthropogenic influences, to explain its significance for aerosol studies. Explicitly identifying specific research gaps will underscore the necessity of the study. Additionally, clearly stating the objectives of study and hypotheses will provide a precise research roadmap. Integrating a comprehensive review of recent, relevant literature will position the research within the broader scientific context, highlighting its novelty and relevance. Does author think that focusing solely on regional measurements can effectively reduce uncertainties

related to aerosol-cloud interactions and radiative forcing? Emphasizing the potential significance and broader impact of the findings, such as advancements in scientific understanding and improvements in climate modeling, will underline the study's importance. Addressing these aspects will enhance the clarity, relevance, and impact of the Introduction, setting a solid foundation for the manuscript.

Although the methodology section comprehensively details the measurements and data analysis, some critical information is missing.

Firstly, the authors need to provide a rationale for selecting the specific periods in summer and winter for the cruise measurements. The inconsistency in measurement periods between summer and winter requires clarification, along with the reasoning behind the chosen cruise routes. This scientific justification is crucial for understanding the relevance to the study's objectives.

Additionally, I recommend incorporating the detailed data quality control procedures (currently in Text S1) into the methodology section, as this is crucial for a measurement report. The authors should clarify why wind direction data from other platforms was not utilized for quality control during winter, particularly in the absence of onboard wind measurements, and explain how ship emissions were prevented from being considered in the measurements. Additionally, the justification for the wind direction ranges used to filter out ship emissions during the summer campaign needs to be provided. Including wind rose plots, similar to those in Hung et al. (2018), would enhance this section. The current data filtering approach, adopted from previous studies, lacks sufficient justification, especially given the differences in cruise measurements and periods. The choice of wind direction ranges and data filtering criteria needs a more logical and scientific basis specific to this study. Furthermore, the rationale for calibrating the ACSM only at the start and end of the cruise, particularly for the longer summer campaign, requires clarification. Additionally, please provide an abbreviation for SMCA. Moreover, The authors mentioned removing abnormal measurement spikes ($> \pm 3\sigma$), attributing them to potential ship emissions. This approach appears arbitrary without concrete reasoning. Providing chemical analyses of particles in these spikes would help determine if they match expected ship exhaust compositions. Additionally, correlating these spikes with wind direction data and specifying their duration, frequency, and handling (whether removed or averaged out) is essential.

Regarding the back trajectory analysis, there is confusion, particularly in winter. Unlike in summer, distinguishing terrestrial and mixed air masses in winter is challenging despite notable differences in particle concentrations and chemical compositions. The short winter cruise period and limited sampling frequency complicate the analysis. To improve clarity, a cluster analysis of back trajectories is recommended to identify distinct air mass origins and pathways, as suggested by Patel et al., 2021, ERC. Including pathway altitudes would enhance understanding of air mass transport. The authors should justify the 48-hour period for back trajectory calculations and consider whether extending this period would provide more comprehensive information. Conducting cluster analysis for distinct source region identification is strongly encouraged.

In several instances, the authors present global statements based on regional studies without

clarifying that the findings are specific to particular regions. For example, the statement in lines 71-72: "Ajith et al. (2022) showed that 64% of particles…" does not universally apply, as the referenced study is not global. It is essential to specify the region where the study was conducted to ensure the accuracy and relevance of the statement. Additionally, numerous instances in the manuscript show incorrect citation of equation and figure numbers, which need to be addressed for clarity and precision.

The manuscript lacks a scientific discussion on how the significantly different chemical compositions in terrestrial and mixed air masses during winter, characterized by high inorganic and low organic concentrations, impact hygroscopicity and CCN activity. It is highly recommended to create a plot showing the contributions of inorganic and organic components for various air masses, further dividing them into specific species (refer to Patel et al., 2021). This approach will provide more comprehensive insights than species plots alone.

Lastly, the discussion on D50 and PNSD for AR and N_CCN calculations lacks clarity and coherence. Begin by clearly explaining the D50 and PNSD methods used for calculating AR and N_CCN, detailing how D50 depends on PNSD. Discuss the variations between these parameters comprehensively. Instead of using a single approach, employ varies methods to calculate hygroscopicity based on D50, based on the chemical composition, considering both internal and external mixtures. Calculate N_CCN and AR accordingly and compare these with observations. Refer to previous studies on CCN closure analysis to provide a clear understanding of whether particle concentrations or chemical composition have a greater impact on CCN activity.

---

## Author Comment (AC1)

We thank the reviewers for taking the time to review our manuscript and for providing constructive advices. We have addressed each of the reviewers' comments individually. Below are our responses to each comment, with the reviewers' comments in black, our responses in red, and the revised manuscript content in italicized orange font.

**#Reviewer 1**

The manuscript presents a study on cloud condensation nuclei (CCN) activity in the South China Sea (SCS), examining seasonal variations and anthropogenic influences through two shipborne campaigns in the summer and winter of 2021. The researchers measured aerosol chemical composition, particle number size distribution (PNSD), and CCN, revealing significant seasonal differences in aerosol properties and the impact of anthropogenic emissions on CCN activity.

Despite these insights, the manuscript appears hastily written, lacking novelty, and presents vague data and discussions that are difficult to interpret. Furthermore, the manuscript fails to clearly differentiate these campaigns from previously reported measurements in the SCS, reducing its contribution to the existing body of knowledge. The manuscript is challenging to follow, primarily describing a lengthy dataset with confusing correlations and writing that diminish readability. The main scientific conclusions are unclear, and the text is overly lengthy, obscuring the scientific merits of the data analysis. It is recommended that the authors focus on a few key scientific findings in their revision. Based on the given title, the manuscript should focus solely on reporting the measurements from the cruise campaigns. However, it also delves into scientific analysis, giving the impression that the authors aim to publish their analytical findings under the guise of reporting measurements. This approach detracts from the clarity and purpose of the study, as the title suggests a primary emphasis on measurement data rather than comprehensive scientific interpretation. In its current form, the manuscript is not suitable for publication in ACP.

Reply: Thanks for the reviewer's comment. According to reviewer's comment, the topic of our article focused on the different effect of anthropogenic influences between summer and winter on CCN activities in SCS. We restructured the manuscript. In the first section of the Results and Discussion (3.1 CCN concentration and aerosol characteristics over SCS in summer and winter), we introduce the basic information of the two cruises conducted in the South China Sea during summer and winter. The second section focuses on the impact of different terrestrial air mass sources on CCN activity in the two seasons (3.2 Anthropogenic influence on CCN concentration in different season). In the last section, we followed reviewer's advice employing the CCN closure method to discuss the effects of aerosol composition and mixing state on CCN activity (3.3 CCN closure analysis).

**Major comments:**

1.Regarding the main scientific findings presented in the manuscript, the discussion does not adequately support these claims. The authors assert that they represent the seasonal variation of aerosol-CCN activity; however, this is not convincingly demonstrated. The winter observations span only 10 days and, unlike the summer campaign, do not spatially cover the entire region. Furthermore, there is no discussion about the potential bias introduced by the limited sampling, which could significantly affect the derived results. Addressing this gap is crucial for validating the study's conclusions.

Reply: Thanks for reviewer's comment. Both the summer and winter cruises are extensive observational missions covering fields such as marine geology, oceanography, and atmospheric science. As a result, the timing and routes of these cruises are planned with an interdisciplinary perspective. The winter cruise has a smaller range and shorter duration (only 10 days) than the summer cruise. This limitation is due to adverse weather conditions, such as strong winter monsoon winds causing poor sea conditions, and the short expedition which considered that it was the first scientific deployment of the research vessel Sun Yat-sen University (lines 145-148).

*Unfortunately, due to adverse weather conditions, such as strong winter monsoon winds causing poor sea conditions, and the fact that it was the first scientific deployment of the research vessel Sun Yat-sen University, the winter cruise had a shorter duration and covered a narrower spatial range compared to the summer cruise.*

In our revised version, we focus on the influence of terrestrial air masses from different sources on CCN activity during the summer and winter seasons. The SCS, characterized by a typical monsoon climate, is predominantly affected by the southwest monsoon in summer and the northeast monsoon in winter. As a result, the terrestrial air masses affecting the South China Sea in winter primarily originate from the Chinese mainland, whereas in summer, they mainly come from the Indochinese Peninsula and the Philippine Islands. Although our winter observations mainly focused on the northern SCS compared to the broader summer observations, they still provide valuable insights into the impact of winter terrestrial air masses on this region.

In the Introduction section, we have added background information on the climate of the South China Sea (Lines 109-122):

*The SCS experiences a typical monsoon climate with distinct seasonal wind direction changes (Wang et al., 2009). The northeast monsoon, occurring from November to March, is characterized by stronger average wind speeds and longer period compared to the southwest monsoon, which dominates from June to August. The transitional periods occur from April to May and September to October. During the northeast monsoon, air pollutants are primarily transported to the SCS by terrestrial air masses from China (Xiao et al., 2017; Liu et al., 2014; Geng et al., 2019). In contrast, during the summer, pollutants mainly originate from terrestrial air masses from the Indochinese Peninsula and Maritime Southeast Asia (Geng et al., 2019; Liang et al., 2021; Sun et al., 2023). These varying sources of anthropogenic emissions exerts different impacts on CCN activity differently across seasons. Additionally, the high cloud fraction over the SCS varies from approximately 0.3 to 0.7 across different months, indicating that aerosol-cloud interactions in the region may differ between seasons (Lu et al., 2022). However, due to limited observational data, our understanding of seasonal variations in CCN activity in the SCS remains incomplete. Conducting comprehensive observational studies on CCN activity across different seasons is essential for improving our understanding of aerosol-cloud interactions in the SCS.*

We agreed that the winter cruise's temporal and spatial limitations present certain constraints to our study. Firstly, the spatial limitations hinder our ability to accurately observe the impact of air masses from the Mainland China on the remote SCS. Secondly, the combined temporal and spatial constraints suggest that other terrestrial air masses, such as those from the Indochinese Peninsula, may also influence the remote SCS. Additionally, we discuss the limitations of our observations in lines 434 to 437:

*Additionally, our observation in winter focused on the CCN activity over the northern SCS, while the influence of air masses from Mainland China in remote SCS was still unclear. Further observations in remote SCS areas could help clarify the anthropogenic*

*influence during winter under the effect of the winter monsoon.*

2. While the title effectively highlights the focus on seasonal variations in CCN activity in the South China Sea (SCS), the Introduction needs further development for full contextualization. It should provide detailed background on the SCS's unique meteorological and environmental characteristics, such as meteorology and anthropogenic influences, to explain its significance for aerosol studies. Explicitly identifying specific research gaps will underscore the necessity of the study. Additionally, clearly stating the objectives of study and hypotheses will provide a precise research roadmap. Integrating a comprehensive review of recent, relevant literature will position the research within the broader scientific context, highlighting its novelty and relevance. Does author think that focusing solely on regional measurements can effectively reduce uncertainties related to aerosol-cloud interactions and radiative forcing? Emphasizing the potential significance and broader impact of the findings, such as advancements in scientific understanding and improvements in climate modeling, will underline the study's importance. Addressing these aspects will enhance the clarity, relevance, and impact of the Introduction, setting a solid foundation for the manuscript.

Reply: We appreciate the reviewer for this valuable suggestion. To clarify our research gap, we have added background information on the climate of the South China Sea and discussed the different impacts of terrestrial air masses under the influence of various monsoons in the Introduction section (Lines 110-123):

*The SCS experiences a typical monsoon climate with distinct seasonal wind direction changes (Wang et al., 2009). The northeast monsoon, occurring from November to March, is characterized by stronger average wind speeds and longer period compared to the southwest monsoon, which dominates from June to August. The transitional periods occur from April to May and September to October. During the northeast monsoon, air pollutants are primarily transported to the SCS by terrestrial air masses from China (Xiao et al., 2017; Liu et al., 2014; Geng et al., 2019). In contrast, during the summer, pollutants mainly originate from terrestrial air masses from the Indochinese Peninsula and Maritime Southeast Asia (Geng et al., 2019; Liang et al., 2021; Sun et al., 2023). These varying sources of anthropogenic emissions exerts different impacts on CCN activity differently across seasons. Additionally, the high cloud fraction over the SCS varies from approximately 0.3 to 0.7 across different months, indicating that aerosol-cloud interactions in the region may differ between seasons (Lu et al., 2022). However, due to limited observational data, our understanding of seasonal variations in CCN activity in the SCS remains incomplete. Conducting comprehensive observational studies on CCN activity across different seasons is essential for improving our understanding of aerosol-cloud interactions on the SCS.*

This emphasizes the purpose for our study, which focuses on the influence of terrestrial air masses on CCN activity in the South China Sea during the winter and summer seasons under the respective monsoon influences. Besides, we stated the focus of our study at the end of the introduction (lines 126-128).

*Our results provide valuable insights into the differences in CCN activity between winter and summer, as well as the influence of different types of terrestrial air masses on CCN activity in the SCS across different seasons.*

3. Although the methodology section comprehensively details the measurements and data analysis, some critical information is missing.

Firstly, the authors need to provide a rationale for selecting the specific periods in summer and winter for the cruise measurements. The inconsistency in measurement

periods between summer and winter requires clarification, along with the reasoning behind the chosen cruise routes. This scientific justification is crucial for understanding the relevance to the study's objectives.

Additionally, I recommend incorporating the detailed data quality control procedures (currently in Text S1) into the methodology section, as this is crucial for a measurement report. The authors should clarify why wind direction data from other platforms was not utilized for quality control during winter, particularly in the absence of onboard wind measurements, and explain how ship emissions were prevented from being considered in the measurements. Additionally, the justification for the wind direction ranges used to filter out ship emissions during the summer campaign needs to be provided. Including wind rose plots, similar to those in Hung et al. (2018), would enhance this section. The current data filtering approach, adopted from previous studies, lacks sufficient justification, especially given the differences in cruise measurements and periods. The choice of wind direction ranges and data filtering criteria needs a more logical and scientific basis specific to this study. Furthermore, the rationale for calibrating the ACSM only at the start and end of the cruise, particularly for the longer summer campaign, requires clarification. Additionally, please provide an abbreviation for SMCA. Moreover, The authors mentioned removing abnormal measurement spikes ($> \pm 3\,\sigma$), attributing them to potential ship emissions. This approach appears arbitrary without concrete reasoning. Providing chemical analyses of particles in these spikes would help determine if they match expected ship exhaust compositions.

Additionally, correlating these spikes with wind direction data and specifying their duration, frequency, and handling (whether removed or averaged out) is essential.

Reply: Thanks for the reviewer's suggestion. Our two cruises were conducted during the prevailing summer monsoon and winter monsoon periods, respectively. We have added the objectives of these two cruises in lines 133-136:

*These two cruises were interdisciplinary scientific expeditions, integrating fields such as marine geology, oceanography, and atmospheric environment. The primary objective in atmospheric environment was to investigate the impact of different monsoons on the atmospheric environment of the South China Sea (SCS).*

Due to the fact that the December 2021 cruise was the first observation mission conducted by the "Sun Yat-sen University" vessel, the meteorological station installed from December 19 to December 22 was under calibration, resulting in a lack of meteorological data. Consequently, in the previous version, we did not use meteorological data for data screening for this cruise. In the revised version, we have included a screening criterion based on relative wind direction and relative wind speed for the period after December 22 in winter. Referring to other literature (Huang et al., 2018; Cai et al., 2020; Liang et al., 2021), we adopted two criteria for screening: the first is based on the components associated with ship emissions such as organic matter, black carbon, and fine particles—if these components showed sudden peaks, they were considered influenced by ship emissions; the second criterion is based on relative wind direction and relative wind speed. We have added a new section in the methods part to describe the data exclusion method in detail (lines 252-266):

*To ensure reliable atmospheric samples in the SCS and mitigate the influence of research vessel emissions, we applied the following data processing procedures (Huang et al., 2018; Cai et al., 2020; Liang et al., 2021).*

*Firstly, we identified organic compounds, black carbon (BC), and small particulate matter (41.4 nm particles) as indicators of ship emissions, recognizing their sudden peak values as indicative of the ship's own emissions.*

*Secondly, we accounted for the relative positions of the ship's chimney and the sampling tube. During the summer cruise, we excluded data corresponding to a relative wind direction (with respect to the ship's bow) between 150° and 270° and a relative wind speed (with respect to the ship's speed) of less than 2.5 m s⁻¹ (Fig. S4a, Fig. S5a1, and Fig. S6a-c). During the winter cruise, we excluded data for a relative wind direction between 150° and 220° and a relative wind speed of less than 2.5 m s⁻¹ (Fig. S4b, Fig. S5b1, and Fig. S6d-f).*

*Applying these criteria, 74.8% of the data in summer and 92.2% in winter (both at 10-minute resolution) were classified as "clean" and retained for analysis. The timeseries of data before and after quality control is shown in Fig. S7.*

We have provided the criteria for screening based on relative wind direction and relative wind speed in the supplement, along with wind rose diagrams for the relevant substances and time series graphs before and after data exclusion.

[Figure]

Figure S4. Instrument and ship chimney location in two cruises.

[Figure]

Figure S5. Wind rose of the relative wind direction (with respect to the bow) and relative wind speed (with respect to the ship speed) in summer and winter cruises; The radius represents the frequency of wind direction occurrences, and the shaded areas indicate wind speed (a1) and (b1); Wind rose of the wind direction and wind speed in summer and winter (a2) and (b2).

**Summer**

[Figure]

Figure S6. Wind rose of the organic, particle in 41.4 nm, and black carbon (BC) in summer (a-c) and winter (c-e) measurements; The radius represents the organic and BC mass concentration and number concentration (dN/dlogDp) of particle in 41.4 nm, and the color indicate wind speed.

[Figure]

Figure S7. Timeseries of particle number size distribution (a) and (e), mass

concentration of NR-PM$_1$ (b) and (f), particle number concentration in 14.6, 41.4, and 109.4 nm (c) and (g), mass concentration of black carbon (d) and (h); The figure number from (a) to (d) means the data in summer, and the figure nnmber from (e) to (h) means the data in winter; The number 1 represented the data before data quality control and the number 2 represent the data after data quality control.

Reference:

1.  Huang, S., Wu, Z. J., Poulain, L., van Pinxteren, M., Merkel, M., Assmann, D., Herrmann, H., and Wiedensohler, A.: Source apportionment of the organic aerosol over the Atlantic Ocean from 53 degrees N to 53 degrees S: significant contributions from marine emissions and long-range transport, Atmos. Chem. Phys., 18, 18043-18062,doi: 10.5194/acp-18-18043-2018, 2018

2.  Cai, M. F., Liang, B. L., Sun, Q. B., Zhou, S. Z., Chen, X. Y., Yuan, B., Shao, M., Tan, H. B., and Zhao, J.: Effects of continental emissions on cloud condensation nuclei (CCN) activity in the northern South China Sea during summertime 2018, Atmos. Chem. Phys., 20, 9153-9167, doi:https://doi.org/10.5194/acp-20-9153-2020, 2020.

3.  Liang, B., Cai, M., Sun, Q., Zhou, S., and Zhao, J.: Source apportionment of marine atmospheric aerosols in northern South China Sea during summertime 2018, Environ. Pollut, 289, 117948, doi:https://doi.org/10.1016/j.envpol.2021.117948, 2021.

4. Regarding the back trajectory analysis, there is confusion, particularly in winter. Unlike in summer, distinguishing terrestrial and mixed air masses in winter is challenging despite notable differences in particle concentrations and chemical compositions. The short winter cruise period and limited sampling frequency complicate the analysis. To improve clarity, a cluster analysis of back trajectories is recommended to identify distinct air mass origins and pathways, as suggested by Patel et al., 2021, ERC. Including pathway altitudes would enhance understanding of air mass transport. The authors should justify the 48-hour period for back trajectory calculations and consider whether extending this period would provide more comprehensive information. Conducting cluster analysis for distinct source region identification is strongly encouraged.

Reply: Thank you for the suggestion. We selected midpoints of the ship trajectories for two periods in summer and winter, and conducted 72-hour back trajectories at 500m hourly, followed by cluster analysis. During the summer cruise, we conducted cluster analysis at two key locations: the midpoint of the ship's track before the outbreak of the summer monsoon (May 5-23) and the midpoint of the track after the summer monsoon began (May 24-June 9). In the winter cruise, cluster analysis was performed at two specific locations: the ship's anchorage near Big Ten-thousand Mountain Island (December 19-22 and December 27-29) and the midpoint between Dawan Mountain Island and Yongxing Island (December 23-26). In summer, we identified three periods: those influenced by terrestrial air masses from Luzon Island ("Luzon" period), from the Indochinese Peninsula ("Indochinese Peninsula" period), and by marine air masses ("Marine-s" period). Due to the small fraction of air masses from Palawan Island, we did not consider them in this study. In winter, we identified periods influenced by terrestrial air masses from Mainland China ("Mainland China" period), Mainland China-marine mixed air masses ("Mixed" period), and marine air masses (Marine-w period).

The result of cluster analysis result is shown in Fig. 4.

[Figure]

Figure 4. The cluster analysis result in summer (a), and winter (b). The solid line in summer means cluster analysis from May 5 to May 24 and the dash line in summer means cluster analysis from May 25 to June 9; The solid line in winter means cluster analysis from Dec 19 to Dec 21 and Dec 27 to Dec 29, and the dash line in winter means cluster analysis from Dec 22 to Dec 26.

5. In several instances, the authors present global statements based on regional studies without clarifying that the findings are specific to particular regions. For example, the statement in lines 71-72: "Ajith et al. (2022) showed that 64% of particles⋯" does not universally apply, as the referenced study is not global. It is essential to specify the region where the study was conducted to ensure the accuracy and relevance of the statement.

Additionally, numerous instances in the manuscript show incorrect citation of equation and figure numbers, which need to be addressed for clarity and precision.

Reply: We have specified the regions in this sentence (lines 71-73):

*Ajith et al. (2022) showed that 64% of particles can be activated as CCN when $\kappa$ is equal to 0.37, whereas when $\kappa$ decreases to 0.23, only 48% of particles can be activated in the tropical coastal area.*

Additionally, we have checked the figure and equation numbers throughout the manuscript to ensure correct referencing.

6. The manuscript lacks a scientific discussion on how the significantly different chemical compositions in terrestrial and mixed air masses during winter, characterized by high inorganic and low organic concentrations, impact hygroscopicity and CCN activity. It is highly recommended to create a plot showing the contributions of inorganic and organic components for various air masses, further dividing them into specific species (refer to Patel et al., 2021). This approach will provide more comprehensive insights than species plots alone.

Reply: Thanks for reviewer's valuable suggestion. We have replotted the figure and present in Fig. 5:

[Figure]

Figure 5. The fraction of NR-PM$_1$ in "Luzon" period (a), "Indochinese Peninsula" period (b), and "Marine-s" period (c) in summer. The fraction of NR-PM1 in "Mainland China" period (d), "Mixed" period (e), and "Marine-w" period (f) in winter.

7. Lastly, the discussion on D50 and PNSD for AR and N_CCN calculations lacks clarity and coherence. Begin by clearly explaining the D50 and PNSD methods used for calculating AR and N_CCN, detailing how D50 depends on PNSD. Discuss the variations between these parameters comprehensively. Instead of using a single approach, employ varies methods to calculate hygroscopicity based on D50, based on the chemical composition, considering both internal and external mixtures. Calculate N_CCN and AR accordingly and compare these with observations. Refer to previous studies on CCN closure analysis to provide a clear understanding of whether particle concentrations or chemical composition have a greater impact on CCN activity.

Reply: We appreciated reviewer's useful advice. In the revision, we use CCN closure method considering aerosol composition and mixing state instead of our origin method to explore the influence of aerosol hygroscopicity in CCN activity (lines 358-388):

*CCN closure study was widely applied to investigate the impacts of different factors on the CCN activity (Patel et al., 2021; Cai et al., 2018; Meng et al., 2014; Deng et al., 2013). In this study, two schemes considering aerosol composition and mixing state based on CCN closure method mentioned in 2.2.3 were applied. The fitting parameter and coefficient of determination ($R^2$) was shown in Table 3 and the fitting plots from two schemes were shown in Fig. S8 and Fig. S9. Besides, the NMB from these two schemes was presented in Fig. 8.*

*In summer, the NMB always lower than 0, which indicated that simulated aerosol hygroscopicity was lower than observed value (Fig. 8). Sea salt which cannot be detected by the ToF-ACSM may account for higher fraction in summer due to low aerosol concentration in summer (Fig. 3c), resulting in the underestimation of aerosol hygroscopicity. The NMB exhibits different trends with changes in SS in "Luzon" and "Indochinese Peninsula" period. Better fitting result appeared in high SS in "Indochinese Peninsula" period, while it appeared in low SS in "Luzon" period (Fig. 8),*

which indicated that aerosol fraction had different trend as particle size increased in these two periods. Besides, "Internal-mixed" scheme had more precious result than it in "External-mixed" scheme in summer (Fig. 8), suggesting the aerosol was primary internally mixed in summer.

In winter, the "External-mixed" scheme always showed a better result than "Internal-mixed" scheme at high SS (0.4% SS and 0.7% SS), indicating that particles in small size were mainly externally mixed. Considering the low hygroscopicity of small-sized particles in winter, it is likely that a significant fraction of these particles consists of externally mixed BC, which probably originated from fresh anthropogenic emissions and remains unmixed with other inorganic salts and organics. As BC ages, inorganic and organic components adhere to it, which would lead to the increase of diameter and particles tended to be internally mixed (Sarangi et al., 2019). This transition resulted in higher hygroscopicity in large-sized particle compared to the smaller-sized particles. Besides, overestimation of aerosol hygroscopicity at high SS could be owing to a higher fraction of non- or less- hygroscopic component (such as organic and BC) at small particle sizes. The predicted $N_{CCN}$ at 0.1% SS are 10%-20% lower than the observed concentrations, whereas the predicted value at 0.2% SS more closely aligns with the observed concentrations (Fig. 8). It could be owing to the higher fraction of sea salt at larger particle size. However, due to instrument limitations, black carbon and sea salt cannot be detected by the ToF-ACSM. More observations containing sea salt and black carbon are needed in the future to better assess their effects on aerosol hygroscopicity in SCS. In addition, further study size-resolved aerosol composition can also enhance the understanding on CCN activity in the SCS.

---

## Author Comment (AC2)

We sincerely appreciate the reviewers for taking the time to review our manuscript and for providing valuable comments. Based on the reviewers' suggestions, we have made revisions to the paper. Below are our responses to each of the reviewers' comments, with the reviewers' comments in black, our responses in red, and the revised manuscript content in italicized orange font.

This manuscript investigates the aerosol and CCN properties over the South China Sea area. The manuscript generally provides some interesting results regarding the seasonal variation and the anthropogenic influences in this area. However, the writing of the manuscript needs a lot of improvements. The focus of the manuscript may also need some changes. Below are some suggestions:

1. I find it very distracting when results from this study are mixed with a lot of results from previous studies in Section 3. It is very difficult to get the point of this study when it is mixed with other studies. For example, lines 227-234, some measurements of particle number concentration are listed. It actually doesn't mean much if only to compare which one is higher and which one is lower. I would suggest adding a table to summarize these measurements so that it is a lot easier for the readers to see which one is higher and which one is lower. Alternatively, results from previous studies can be organized into the Introduction section so that the readers can have a better background from the beginning. The comparison between the previous results and this study can also be organized into a new Discussion section. Basically, I would prefer that Section 3 focuses on the results only from this study. Similarly, lines 236-238, lines 249-251, lines 256-265, lines 298-308, lines 321-323, lines 344-345, lines 362-367, lines 374-377, lines 380-388, and lines 392-396 are all results from previous studies. Please put these previous results in a table, or put them into Introduction or a new Discussion section. This would help the readers to integrate the results from this study.

Reply: We have rewritten the first section of the results to focus specifically on the content of our study (lines 269-299):

[revised manuscript text omitted]

2. It is very interesting to see that aerosol and CCN properties are quite different in summer and winter. This indicates that the seasonal variations of aerosol and CCN properties should be considered in regional or climate models when studying aerosol-cloud interaction for this area. I especially agree that particle composition and kappa are quite different in summer and winter. However, it is not very clear if particle number concentration is significantly different in summer and winter. The manuscript emphasizes that summer has much higher number concentration when the marine atmosphere is influenced by terrestrial air masses (lines 226-227). I agree. But how about the average number concentration in the whole observed period in summer? Is it still much higher than that in winter? For many days when the atmosphere is influenced by mixed air masses, particle number concentration seems to be similar in summer and winter. So please check if the number concentration in the whole summer period is on average much higher than that in winter. The panel d in Figure 2 cannot provide such information because summer and winter are not plotted with the same scale. It would help if the properties are plotted in the same scale for summer and winter in Figure 2.

In addition, regarding the dominant mode in PNSD, it is said in line 30 that PNSD has a dominance of Aitken mode in summer and a dominance of accumulation mode in winter. But based on Figure 2 and Figure 3, I would say that both Aitken mode and accumulation mode are important in winter. It is not appropriate to conclude that the dominant mode in winter is accumulation mode. So I would see more evidence regarding the dominant mode. Maybe you could calculate the total number concentration in Aitken mode and in accumulation mode and compare to see which one is dominant.

Reply: Thanks for reviewer's comment. The total particle number concentration was higher in summer (6966 cm$^{-3}$) than in winter (4988 cm$^{-3}$) (Table 1). We have replotted the Figure 2 to make sure the particle number concentration value is in the same scale. Besides, the Aitken mode particle concentration was similar to accumulation mode particle in winter, so we revised our description in lines 278-280:

*In summer, particles were concentrated in smaller sizes, whereas in winter, particle size distribution was relatively balanced between the Aitken mode (2185 cm$^{-3}$) and the accumulation mode (2176 cm$^{-3}$) (Fig. 3a-b).*

[Figure]

Figure 2. Timeseries of (a) particle number size distribution, (b) mass concentration of NR-PM$_1$, and (c) its fraction, (d) mass concentration of organic carbon and elemental carbon, (e) number concentration of total particle and cloud condensation nuclei under the supersaturation of 0.1%, 0.2%, 0.4%, and 0.7%, and (f) aerosol hygroscopicity. The number 1 in figure number means timeseries in summer and number 2 means it in winter.

3. I think the size-resolved AR should be shown, especially because D50 is determined by fitting the AR and dry diameter at each supersaturation.

Reply: We have added the figure of size-resolved AR fitting result as Fig. S3.

[Figure]

Figure S3. The average size-resolved activation ratio (AR) fitting result at 0.2% SS (a), 0.4% SS (b), and 0.7% SS (c) in summer; The average size-resolved activation ratio (AR) fitting result at 0.1% SS (d), 0.2% SS (e), 0.4% SS (f), and 0.7% SS (g) in winter.

4. The method in 2.2.1 (equation 1) and 2.2.2 (equation 4) seem to be in conflict. In equation 1, I assume that the activation ratio is size-resolved? However, in Equation 4, the activation ratio is the bulk activation ratio? So please clarify whether AR represents the size-resolved or the bulk activation ratio.

Reply: We obtained the activation diameter ($D_{50}$) from size-resolved AR and diameter according to SMCA method. The CCN concentration ($N_{CCN}$) was calculated based on the $D_{50}$ and observed PNSD from SMPS. Then the AR represent bulk AR from ratio of CCN concentration to total particle concentration. We have added the following sentence in lines 188 and 220 for clarification:

*where AR is the size-resolved AR* (line 192)

*It is noting that the AR here is bulk AR.* (line 224)

5. Regarding the size-dependent kappa value in lines 282-283, please clarify the reasons why kappa value is size-dependent.

Reply: To clearly express our observational results, we have revised our statement to indicate that hygroscopicity varies with changes in supersaturation (SS) (lines 293-295):

*The hygroscopicity pattern varied between seasons: in summer, κ increased with SS (from 0.49 to 0.72 between 0.2% SS and 0.4% SS), while in winter, κ decreased with SS (from 0.50 to 0.15 between 0.1% SS and 0.7% SS) (Fig. 3a-b).*

And we discuss the reason why kappa value is size-dependent in the following section. The possible reason for higher hygroscopicity at high SS in summer than winter is the MSA oxidized from DMS produced by phytoplankton (lines 341-345):

*However, aerosol hygroscopicity at small sizes was much lower in the "Mainland China" period than in the "Luzon" period (Fig. 8), contributing to the low AR in the "Mainland China" period (Fig. 7). This lower hygroscopicity could be due to lower sulfate concentration, oxidized by DMS, in winter than in summer, as higher sea surface*

*temperatures in summer (29.3°C) compared to winter (18.0°C) promote DMS production by phytoplankton (Bates et al., 1987).*

Besides, Additionally, the low hygroscopicity of small particles in winter may be due to externally mixed BC and an increased proportion of organics. (lines 373-382):

*In winter, the "External-mixed" scheme always showed a better result than "Internal-mixed" scheme at high SS (0.4% SS and 0.7% SS), indicating that particles in small size were mainly externally mixed. Considering the low hygroscopicity of small-sized particles in winter, it is likely that a significant fraction of these particles consists of externally mixed BC, which probably originated from fresh anthropogenic emissions and remains unmixed with other inorganic salts and organics. As BC ages, inorganic and organic components adhere to it, which would lead to the increase of diameter and particles tended to be internally mixed (Sarangi et al., 2019). This transition resulted in higher hygroscopicity in large-sized particle compared to the smaller-sized particles. Besides, overestimation of aerosol hygroscopicity at high SS could be owing to a higher fraction of non- or less- hygroscopic component (such as organic and BC) at small particle sizes.*

6. Regarding the AR ratio in lines 390-391 in Section 3.3, what does "beyond D50" mean? The meaning of this sentence is not clear. Please revise. This is very important for understanding the sensitivity tests afterwards. I think it's quite interesting that PNSD is the most important factor influencing Nccn in summer, whereas hygroscopicity is the most important factor in winter (lines 416-417). This is based on the sensitivity tests performed in this study. I wonder if the authors can provide any underlying physical reasons for this.

Reply: Thanks for reviewer's question. In reference to the two reviewer's comments and other literatures, we consider that the original method, based on sensitivity experiments, may not accurately explain the effects of PNSD and hygroscopicity on CCN concentrations. Therefore, we removed the original method in revised version and apply CCN closure method, which has been widely used in other researches and can provides more accurately result, to analyze the impact of aerosol composition and mixing state on CCN activities. The CCN closure analysis was shown in 3.3 (lines 358-388).

*CCN closure study was widely applied to investigate the impacts of different factors on the CCN activity (Patel et al., 2021; Cai et al., 2018; Meng et al., 2014; Deng et al., 2013). In this study, two schemes considering aerosol composition and mixing state based on CCN closure method mentioned in 2.2.3 were applied. The fitting parameter and coefficient of determination ($R^2$) was shown in Table 3 and the fitting plots from two schemes were shown in Fig. S8 and Fig. S9. Besides, the NMB from these two schemes was presented in Fig. 8.*

*In summer, the NMB always lower than 0, which indicated that simulated aerosol hygroscopicity was lower than observed value (Fig. 8). Sea salt which cannot be detected by the ToF-ACSM may account for higher fraction in summer due to low aerosol concentration in summer (Fig. 3c), resulting in the underestimation of aerosol hygroscopicity. The NMB exhibits different trends with changes in SS in "Luzon" and "Indochinese Peninsula" period. Better fitting result appeared in high SS in "Indochinese Peninsula" period, while it appeared in low SS in "Luzon" period (Fig. 8),*

*which indicated that aerosol fraction had different trend as particle size increased in these two periods. Besides, "Internal-mixed" scheme had more precious result than it in "External-mixed" scheme in summer (Fig. 8), suggesting the aerosol was primary internally mixed in summer.*

*In winter, the "External-mixed" scheme always showed a better result than "Internal-mixed" scheme at high SS (0.4% SS and 0.7% SS), indicating that particles in small size were mainly externally mixed. Considering the low hygroscopicity of small-sized particles in winter, it is likely that a significant fraction of these particles consists of externally mixed BC, which probably originated from fresh anthropogenic emissions and remains unmixed with other inorganic salts and organics. As BC ages, inorganic and organic components adhere to it, which would lead to the increase of diameter and particles tended to be internally mixed (Sarangi et al., 2019). This transition resulted in higher hygroscopicity in large-sized particle compared to the smaller-sized particles. Besides, overestimation of aerosol hygroscopicity at high SS could be owing to a higher fraction of non- or less- hygroscopic component (such as organic and BC) at small particle sizes. The predicted $N_{CCN}$ at 0.1% SS are 10%-20% lower than the observed concentrations, whereas the predicted value at 0.2% SS more closely aligns with the observed concentrations (Fig. 8). It could be owing to the higher fraction of sea salt at larger particle size. However, due to instrument limitations, black carbon and sea salt cannot be detected by the ToF-ACSM. More observations containing sea salt and black carbon are needed in the future to better assess their effects on aerosol hygroscopicity in SCS. In addition, further study size-resolved aerosol composition can also enhance the understanding on CCN activity in the SCS.*

7. Section 3.4 seems to have conflicting results with Section 3.3. It is shown in Section 3.3 that PNSD determines Nccn in summer and hygroscopicity determines Nccn in winter. However, in Section 3.4, it is shown that PNSD is important for Nccn in both summer and winter. I think Section 3.3 and Section 3.4 should be better integrated. The writing should be more concise and focused.

Reply: Thanks for reviewer's question. Firstly, we consider that this method, based on sensitivity experiments, may not provide accurate explanation on the effects of PNSD and hygroscopicity on CCN concentrations. Therefore, we adopted the more widely used CCN closure method as a replacement. Additionally, we reconsidered the influence of terrestrial air masses in summer and winter. Cluster analysis revealed two distinct continental air masses in summer: one from the direction of Luzon Island and the other from the Indochinese Peninsula. To provide clearer and more comprehensible results, we integrated Section 3.4 into Sections 3.2 and 3.3.

Minor points:

1. Please provide a little discussion on cloud climatology for the studied area. It would be good to see that there is a relatively high cloud fraction, especially warm cloud in the studied area.

Reply: To our current knowledge, there is still a lack of research on the warm cloud on the SCS. Thus, we introduced the seasonal variation of high cloud fraction in SCS in lines 118-120:

*Additionally, the high cloud fraction over the SCS varies from approximately 0.3 to 0.7 across different months, indicating that aerosol-cloud interactions in the region may differ between seasons (Lu et al., 2022).*

2. Lines 431-433 are very similar to lines 448-450. The writing should be improved.

Reply: We employed a new method, CCN closure analysis, to revise Section 3.3, replacing the original Sections 3.3 and 3.4.

3. The manuscript title and the titles in Section 3 are kind of confusing. A lot of "impact of … on…." Or "influence of … on …" are seen in the titles. For example,

Title of the manuscript: "anthropogenic influence on CCN activation"

Title of 3.2: Impact of chemical composition on hygroscopicity

Title of 3.2.1: impact of inorganic components

Title of 3.2.2: impact of organic components

Title of 3.4: influence of spatial distribution of particle properties on NCCN

In addition, the title of 3.1 is too simple. This title does not provide much information. There are also some inconsistency in the titles. For example, I can see that seasonal variation is a focus of this study based on the title of the manuscript. However, only the title of 3.3 has "seasons". Based on the current titles, it is hard to figure out which part in Section 3 actually discusses "seasonal variations".

Reply: We have changed our title:

3.1 CCN concentration and aerosol characteristics over SCS in summer and winter

3.2 Anthropogenic influence on CCN concentration in different season

3.3 CCN closure analysis

In Section 3.1, we briefly introduce the observation result in summer and winter. The detail discussion about the impact of different types of terrestrial air masses on CCN activities in summer and winter. In the last section, we further discuss the influence of aerosol composition and mixing state on CCN activities in SCS.

4. In section 3, there are some sentences that are repetitive. For example, lines 266-268, and lines 280-282. Section 3 should be integrated in a better way. After the results from previous studies are moved to other places, Section 3 can be more focused and better integrated.

Reply: We have added Table 1 and replotted the Figure 3 to present the differences between this study and other researches. Besides, we have rewritten the 3.1 to focus on presenting the result on this study.

5. It seems the organic carbon and elemental carbon are missing in Figure 2. See the figure caption, (d).

Reply: We have deleted the data of organic carbon and elemental carbon in our discussion and changed the figure caption in Figure 2.

6. Equation 1: Nccn/Ncn should be replaced with AR, to be consistent with the use of AR in Equation 4.

Reply: We have changed Nccn/Ncn to AR.

7. Equation 2: the two formula should be put in separate lines.

Reply: We have put them in separate lines.

8. Line 448: "in winter, the increasing trend…" should be changed to "in winter, the decreasing trend".

Reply: We have deleted this sentence.

9. Line 39: "impact of PNSD on AR was greater than on aerosol hygroscopicity in summer" should be changed to "impact of PNSD on AR was greater than aerosol hygroscopicity in summer". In addition, "vice versa" is not a good expression in this

sentence.

Reply: We have deleted this sentence.

---

## Referee Report (RR1)

**Review of egusphere-2024-956**

This manuscript presents the CCN data obtained from shipborne measurement over the South China Sea. Because CCN measurement is still lacking, and especially measurement over the sea has been even rarely done, presenting CCN data measured over the sea is very valuable. However, this manuscript lacks the integrity and quality of a paper that would be worthy of being published in ACP. First, measurement setup and data analysis methods are not clearly explained. Figures and Tables are not clearly explained in captions and lack some important information. English should be greatly improved. For these reasons, I recommend resubmission of the manuscript after all my comments are addressed properly. Below are my specific comments.

**Major Comments**

Section 2.1.2:

Explain clearly how size-resolved CCN and PNSD were measured simultaneously. Do the authors have an SMPS for PNSD measurement and a separate DMA that can be used to setup a DMA-CCNC system for size resolved CCN? If that is not the case, with one SMPS, how can they measure both? The authors have CCNC-200 that has two CCN measurement columns. So, it could have been possible to use one column for size resolved CCN measurement and another for regular CCN concentration measurement at several SSs. But apparently CCN concentration at a given SS was obtained from the integration of size-resolved CCN data, instead of making direct CCN measurement. I wonder why the two column capability of CCNC-200 was not fully utilized. No clear explanation is given. Relevant to this section is the fitting results in Figure S3, which seem to show the averaged size-resolved activation ratio (AR) for the entire summer and for the entire winter periods, respectively. Since the aerosol characteristics are likely different for different air masses, the size-resolved AR should be estimated for each cluster and then calculate $D_{50}$. Apparently the authors have done that (Fig. 7). Then I wonder how the results of Figure S3 are produced. The authors should clearly explain.

Section 2.2.3:

Why did the authors predict CCN concentration when direct measurement was possible with one of the two columns in CCNC-200? Anyway, later in the manuscript, this "predicted" $N_{CCN}$ was apparently used as $N_{CCN,obs}$, when doing the CCN closure. Is this really the case? Explain clearly.

Section 2.2.4:

Regarding the cluster analysis, which method is used to classify the clusters? Is it a hierarchical clustering method? If so, is it bottom-up approach or top-down approach? If not, is k-means clustering method or fuzzy c-mean clustering method used? The authors should describe the

method clearly.

It is implied that the authors know exactly on what day the summer monsoon started. Can this be so clearly known? If so, explain clearly how so by showing the supporting data (e.g., wind pattern change, …).

Unlike ground (fixed location) measurement, cluster analysis for ship measurement requires some caution since the research vessel is moving (i.e., cruising). To ensure the representativeness and suitability of the midpoint used as the starting location, the back trajectories for the ship's coordinate during the cruises and the back trajectories for the midpoint of the ship track at the same time should be close enough. The authors should confirm if this is the case by showing supporting data.

Section 3.1:

The absolute difference of $N_{CCN}$ between the two seasons was larger at higher SS but that should be natural since $N_{CCN}$ becomes higher at higher SS. The comparison should be relative: the ratio of $N_{CCNwinter}/N_{CCNsummer}$ at a given SS should be shown for such comparison.

The $\kappa$ values shown in Table 1 and Fig.3a do not match for 0.4% SS (0.74 vs. ~0.60). In the text, it is 0.72. A good example of poor sincerity of this manuscript! At 0.2% SS, summer and winter $\kappa$ were 0.49 and 0.31, respectively. Can they be considered "similar" as the authors stated? I do not think so.

It should also be noted that the estimated $\kappa$ values are for the particles of critical diameter ($D_{50}$), the smallest particles that can be activated at a given SS. So, these $\kappa$ values do not represent the $\kappa$ values of all the particles that can be activated at a given SS. This should be stated clearly before any arguments are made on $\kappa$ values.

Section 3.2:

To ensure that cluster analysis is well-conducted, I would suggest that the authors present all the back trajectories classified in each cluster in the supplement.

If a back trajectory does not stay long enough within the boundary layer, it is difficult to say that it reflects the characteristics of the air masses where it passed. Therefore, it is recommended that the altitude of the back trajectory is also presented, to more clearly demonstrate the influence of the specified regions like "Mainland China", "Luzon", and "Indochinese Peninsula" on SCS. This can be confirmed by averaging the altitudes of all backward trajectories in each cluster.

The authors state that low hygroscopicity of 'Mainland China' could be due to low sulfate concentration oxidized by DMS in winter than in summer. Here the comparison is between 'Mainland China' and 'Luzon,' the two terrestrial regions. So I do not understand why DMS production is discussed here, which are definitely the source of CCN over marine regions. Explain the relevance.

**Minor Comments**

Line 117: "different impacts on CCN activity differently across seasons" --> " different impacts on CCN activity across the seasons"; What does "high cloud fraction" mean? Fraction of 'high cloud' or high fraction of clouds? If the former is the case, why is this relevant?

Line 134: "different monsoons" --> "summer and winter monsoons"

Line 148: Add the information of the actual height of the sampling lines from the sea surface.

Lines 172-175: Since no result on OC/EC were discussed in the manuscript, it is inappropriate to mention OC/EC in Section 2.1.3. Likewise, the discussion of trace gases seems not to be presented in the manuscript and/or supplement. It might be worth checking.

Lines 181-182: Writing December 22nd as 12.22 can be confusing to readers. Unify the expression for date throughout the manuscript. Better to be December 22nd or 22 December but not 12.22.

Line 186: 'praticle' --> 'particle'

Line 191: "AR is the size-resolved AR" --> "AR indicates the size-resolved AR value" In several occasions later in the manuscript, AR seems to indicate the bulk AR value. These should be clearly explained.

Line 210: "$\kappa$ from ..." --> "$\kappa$ for ...", 'Nacl' --> 'NaCl' here and for other occasions.

Line 211: $\kappa$ of organic was assumed to be 0.1. Where is it from?

Line 217: Eq. (4) can give predicted CCN concentration under the assumption that all particles of diameter greater than $D_{50}$ activate for the given SS. Where is the justification?

Line 219: "number concentration under specific" --> "number density of specific"

Line 230-233: What does "have identical concentration at each size" mean? Is this inteneded for "fixed proportion for all sizes?" How does $D_{50}$ calculated for each species? Explain clearly.

Line 236: In Eq. (6), $N_{CCN,obs}$ is not an observed value, strictly speaking. Be clear on this.

Line 246: 'outbreak' --> 'onset'

Lines 280-281: Aerosol number concentration is higher in summer than in winter, but mass concentration is higher in winter. Based on Figure 3, however, Aitken mode particles are much more abundant in summer, while accumulation mode particles, which greatly affect mass concentration, are similar between summer and winter. Then, why are arosol mass concentration significantly different between summer and winter? Need more explanation.

Line 296: Add (Cai et al., 2020) after "Guangzhou", just like in the previous sentence.

Line 363: "NMB always lower" --> "NMB was always lower"

Line 386: "study size-resolved" --> "study of size-resolved"

Line 414-415: "higher particle fraction in the accumulation mode compared to" --> "higher fraction of the accumulation mode particles comparted to"

Line 434: In "northern SCS," what does 'northern' exactly mean? The winter cruise route shown in Fig. 1 does not seem to indicate cruising over northern part of SCS. In the same context, what does "remote SCS" mean?

Table 1: Show $D_{50}$ values at least for this study. Is AR a bulk AR value? Explain in the caption. Why 'Northern' for winter cruises? There are several CCN measurement studies over the Yellow Sea, which would represent influence of northern part of continental China and therefore can provide good contrasting results.

Table 2: Add AR values in a separate column and widen the column to show data in one line.

Figure 1: Add important place names and mark the mid-points of back trajectories in (a).

Provide full explanation in the caption.

Figure 4: Show important place names.

Figure 7: Why are there no κ plots for Indochinese Peninsula and Marine for 0.7% SS? Explain in the main text.

---

## Referee Report (RR2)

**Review of 'Measurement Report: Cloud condensation nuclei (CCN) activity in the South China Sea from shipborne observations during summer and winter of 2021: seasonal variation and anthropogenic influence'**

This manuscript presents a comprehensive aerosol and CCN properties over the South China Sea (SCS) during summer and winter from two ship-based measurements. The aerosol size distribution, chemical composition of PM1, hygroscopicity, and the CCN (with closure study) are examined and briefly analyzed, and the seasonal variations in the aerosol species and activation ratio is evident. This study provides extra data sources for future studies over the SCS. I think this manuscript holds the potential of publication, after considering and addressing the concerns and questions I have listed below.

**Measurements**

Please specify the instruments (CCN counter, SMPS, and DMA), measurable size range (and bin size if applicable), frequency, and instrumental uncertainty on the size-resolved number concentrations.

What is the supersaturation ramping time scale for the CCN column A?

Also, the error or precision of the ACSM measured mass concentration should be specified after the composition-dependent collection efficiency correction.

The uncertainties for the meteorological quantities need to be reported.

More details on the aethalometer are also encouraged (size ranges, uncertainty, etc.).

In the CCN closure section, can you report representative D50 values for four species (as in S7) under the External scheme, during summer and winter seasons in the study domain?

I believe those are crucial for a measurement report.

**Specific Comments:**

**Line 117:** Can you elaborate on how the seasonal variation of the fraction of high cloud is relevant here in terms of the ship-observed CCN near the sea surface and the aerosol-high cloud interaction? Intuitively, would the lower boundary layer aerosol/CCN have greater impacts on the MBL clouds, or is there any particular dynamical mechanism the author is referring to?

**L153.** 'The … SMCA… was initially…'. And do you want to say you utilize this sampling strategy to get the size-resolved CCN?

**L159.** Have both sensors in the counter column B on two ships malfunctioned during the two sampling periods?

**L206.** You have introduced D50 as 'particle size at which 50% of the particles are activated at a specific SS' before. Your statement here is flawed.

**L275.** The results in S8 and S9 are well-justified, though, you should consider putting a few words about it in the main text.

**L307.** Winter period shows two peaks with more organic (less sulfate) and with both high organic and sulfate. Elaborate on how the impact of Northeast Monsoon 'persist'.

**L311.** Define clearly how the Nucleation, Aikten, and Accumulation are defined, in terms of size cut and rationale, in this study (should have been done in the Data section).

**L330.** Explanation is needed on the flipped hygroscopicity-supersuration relations between summer and winter.

**L388.** Which figure or table are you referring to wrt. 'smaller sizes'. And yes, sulfate fraction is reduced in winter 'Marine' period, but the increased ammonium may compensate for this effect (Fig. 5f). You may also consider attributing this to the increases in organic aerosol contribution due to factors like reduced photochemical oxidation.

**Technical Comments**

**L56.** '…partially attributed…'. And considering adding more recent and relevant references to these statements.

**L71.** Please be more specific on what particles (compositions, sizes, etc.) were examined in Ajith et al. (2022)

**L90.** This paper (Zheng et al., 2020) is not on the reference list. And I presume you refer to 'Eastern North Atlantic'.

**L153.** 'The … SMCA… was initially…'.

**L153.** '(Fig. 1c2)'

**L377.** Here, the statement, though reasonable, has not been supported by the results, use 'potentially led' instead.

**Figures.** Please put the figure caption directly beneath the figure.

**Fig. 3.** Same y-axis range is needed for (a) and (b). And please state the seasons for (e) and (f).

**Fig. S9.** Panel (c) and (f), consider using something like 'Marine-Win'? I was confused with Marine-(South) and Marine-(West) at first glance.

**Fig. S14.** Use more distinguishable colors between 14.6 and 41.4 nm. And the subpanel labels do not match the captions for winter.

**Reference.** The reference list is hard to distinguish between entries, so please correct the format.

---

## Author Response (AR2)

**#Reviewer 1**

We sincerely appreciate the reviewer's positive feedback and useful suggestion. Based on the reviewers' suggestions, we have revised the paper. Below are our responses to each of the reviewers' comments, with the reviewers' comments in black, our responses in red, and the revised manuscript content in italicized orange font.

Then manuscript has been significantly improved. The content in Section 3 is more focused now. The new figure 3 shows clear difference in summer and winter regarding aerosol size distribution and composition. This result is better integrated in the current version than in the previous version of the manuscript. The new Table 1 also makes the comparison between this study and previous studies more straightforward. But the manuscript still needs some revision:

Reply: Thanks for reviewer's valuable comment. We have modified the Fig. S3, corrected the grammars and made some change in Section 3 in this revision.

1. In the new Figure S3, the fonts are small and not clear. Please make sure the figure is clear in the final version. There is no need to plot the figure on a scale of 2 in the vertical axis, because the AR can only get as high as 1. I think the new Figure S3 can be integrated, with the three figures for the summer plotted in one panel, and the three figures for winter plotted in another panel.

Reply: We appreciate the reviewer for this valuable suggestion. We have increased the font size and added size-resolved information in Figure S3 (now labeled as Figure S5). To avoid confusion from combining information for different supersaturation (SS) with additional size-resolved AR during different periods, we decided to plot subfigures for each SS individually.

[Figure]

*Figure S5. The average size-resolved activation ratio (AR) fitting result at 0.2% SS (a), 0.4% SS (b), and 0.7% SS (c) in different periods in summer; The average size-resolved activation ratio (AR) fitting result at 0.1% SS (d), 0.2% SS (e), 0.4% SS (f), and 0.7% SS (g) in different periods in winter.*

2. The manuscript still needs to be checked for grammar errors.

Reply: We appreciate the reviewer for this valuable suggestion. We have checked our manuscript and corrected the grammar errors.

3. The writing in Section 3 is still not smooth, especially paragraphs 2-6 in Section 3.2. The content is fine. But please revise the writing so that it can read more smoothly.

Reply: We appreciate the reviewer for this valuable suggestion. We have revised the content of Section 3.2 to enhance its readability. Additionally, a native speaker reviewed the text for grammatical accuracy and helped refine the wording.

**#Reviewer 2**

We sincerely appreciate the reviewers for taking the time to review our manuscript and for providing valuable comments. Based on the reviewers' suggestions, we have revised the paper. Below are our responses to each of the reviewers' comments, with the reviewers' comments in black, our responses in red, and the revised manuscript content in italicized orange font.

This manuscript presents the CCN data obtained from shipborne measurement over the South China Sea. Because CCN measurement is still lacking, and especially measurement over the sea has been even rarely done, presenting CCN data measured over the sea is very valuable. However, this manuscript lacks the integrity and quality of a paper that would be worthy of being published in ACP. First, measurement setup and data analysis methods are not clearly explained. Figures and Tables are not clearly explained in captions and lack some important information. English should be greatly improved. For these reasons, I recommend resubmission of the manuscript after all my comments are addressed properly.

Reply: Thank you for the reviewer's insightful comments. In this revised version, we have added specific details about the SMCA method and included additional data to strengthen the reliability of our cluster analysis. We have also enhanced the figures and tables in accordance with the reviewer's suggestions. Furthermore, we thoroughly reviewed the manuscript and enlisted the help of a native English speaker to correct any grammatical errors and improve the overall quality of the writing.

Below are my specific comments.
Major Comments
1. Section 2.1.2: Explain clearly how size-resolved CCN and PNSD were measured simultaneously. Do the authors have an SMPS for PNSD measurement and a separate DMA that can be used to setup a DMA-CCNC system for size resolved CCN? If that is not the case, with one SMPS, how can they measure both? The authors have CCNC-200 that has two CCN measurement columns. So, it could have been possible to use one column for size resolved CCN measurement and another for regular CCN concentration measurement at several SSs. But apparently CCN concentration at a given SS was obtained from the integration of size-resolved CCN data, instead of making direct CCN measurement. I wonder why the two-column capability of CCNC-200 was not fully utilized. No clear explanation is given. Relevant to this section is the fitting results in Figure S3, which seem to show the averaged size-resolved activation ratio (AR) for the entire summer and for the entire winter periods, respectively. Since the aerosol characteristics are likely different for different air masses, the size-resolved AR should be estimated for each cluster and then calculate D50. Apparently, the authors have done that (Fig. 7). Then I wonder how the results of Figure S3 are produced. The authors should clearly explain.

Reply: Thanks for reviewer's comment. Unfortunately, we only have one SMPS and cannot afford another DMA for the size-resolved CCN measurement. Thus, we have to combine the SMPS and CCNc to simultaneously measure the PNSD and size-resolved CCN activity, following the Scanning Mobility CCN Analysis method (Moore et al., 2020). The instrument setup is illustrated in figure S2. Initially, particles were passed through a Nafion dryer to remove moisture and were then neutralized using a neutralizer. The particles were subsequently size-selected using a DMA. Afterward, the particle flow was split between a CPC for particle concentration measurement and a CCNc for

CCN measurement at a specified supersaturation. A dilution air (0.5 LPM) was added to the CPC inlet to maintain the sample flow through the DMA. The effect of the dilution air has been considered during the PNSD data processing. Therefore, we were able to measure both particle concentration and CCN concentration with a single SMPS and CCNc. We have provided additional details of this method in the manuscript (Lines 159-165):

*During the SMCA measurement, the particles were first passed through a Nafion dryer to remove moisture, then neutralized using a neutralizer. After that, they were subjected to size selection with a DMA. The particles were then split between a CPC (1 L min$^{-1}$) for particle concentration measurement and a CCNc (0.5 L min$^{-1}$) for CCN measurement at a specific supersaturation. To maintain sample flow through the DMA, dilution air (0.5 L min$^{-1}$) was added to the CPC inlet stream. The effect of the dilution air was accounted for in the PNSD data processing.*

[Figure]

*Figure S2. Instrument setup of the SMCA system.*

To validate the reliability of the PNSD measurements obtained by using the instruments setup in the SMCA method, we present data from previous observations at the Heshan supersite in the Guangdong Province of China during the fall season 2019 (Cai et al., 2021). These data compare the PNSD measured by the DMA (model 3081A, TSI Inc., USA) and CPC (model 3775, TSI Inc., USA) in the SMCA method with those directly measured by an SMPS (DMA model 3081A and CPC model 3775, TSI Inc., USA). The similar PNSD measured by the two methods, along with the strong correlation in the total particle concentrations obtained (The coefficient of determination ($R^2$) is 0.95), indicate a high level of consistency between the results from these two-measurement method. (Fig. 1.1).

[Figure]

Figure 1.1. Particle number size distribution from SMPS (a), particle number size distribution from DMA and CPC in SMCA method (b), comparison of particle number concentration from SMPS (black line) and from SMCA method (red dash line).

Unfortunately, due to the malfunction of flow sensor in the column B, the data from this column was unavailable during these two measurements. We could only use the A column for observations, which prevented us from directly measuring the total $N_{CCN}$. In the manuscript, we have thoroughly explained the rationale behind using only one column for our analysis (Lines 158-159):
*Unfortunately, due to the malfunction of flow sensor in the column B, only the data from column A is presented in this study.*

The activation ratio presented in the Figure S3 originally represented the average size-resolved AR for both summer and winter. To provide a clearer depiction of the size-resolved AR curves across various periods, we have revised and redrawn the figure (now labeled as Figure S5).

[Figure]

*Figure S5. The average size-resolved activation ratio (AR) fitting result at different SS during various periods in summer (a, b, c) and winter (d, e, f, g).*

Reference:
Moore, R. H., Nenes, A., and Medina, J.: Scanning Mobility CCN Analysis-A Method for Fast Measurements of Size-Resolved CCN Distributions and Activation Kinetics, Aerosol Sci Tech, 44, 861-871, doi: https://doi.org/10.1080/02786826.2010.498715, 2010.
Cai, M., Liang, B., Sun, Q., Liu, L., Yuan, B., Shao, M., Huang, S., Peng, Y., Wang, Z., Tan, H., Li, F., Xu, H., Chen, D., and Zhao, J.: The important roles of surface tension and growth rate in the contribution of new particle formation (NPF) to cloud condensation nuclei (CCN) number concentration: evidence from field measurements in southern China, Atmos. Chem. Phys., 21, 8575-8592, doi:https://doi.org/10.5194/acp-21-8575-2021, 2021.

2. Section 2.2.3: Why did the authors predict CCN concentration when direct measurement was possible with one of the two columns in CCNC-200? Anyway, later in the manuscript, this "predicted" $N_{CCN}$ was apparently used as $N_{CCN,obs}$, when doing the CCN closure. Is this really the case? Explain clearly.

Reply: We appreciate the reviewer for this valuable suggestion. Unfortunately, flow sensor in the column B malfunctioned. We derive the total CCN concentration from the observed size-resolved AR and particle number concentration. In the revision, we recalculated total CCN concentration by integrating the size-resolved AR curves with the actual particle concentrations for improved accuracy.

$$N_{CCN}(SS) = \int_0^\infty AR(SS, D_P)N_{CN}(D_P)dD_p$$

where $N_{CCN}$ (SS) is CCN concentration at a specific SS, AR (SS, $D_p$) is the AR on a certain diameter at a specific SS from the SMCA method and $N_{CN}(D_P)$ is the particle number density of specific diameter from SMPS measurement.

Previous researches have shown that this method (size-resolved CCN from one column in CCNc-200) provides results closely matching those obtained from direct measurement (from another column in CCNc-200), supporting its reliability (Meng et al., 2014; Lathem and Nenes, 2011). Consequently, we refer to the CCN concentration derived using this method as the observed CCN concentration. We have reintroduced the details of the calculation in the manuscript and explained the rationale for referring

to the CCN concentration obtained using this method as the observed CCN concentration (Lines 222-236):

*Due to the malfunction of flow sensor in the column B, the CCN concentration ($N_{CCN}$) can be calculated based on the size-resolved AR at a specific SS from SMCA method and observed particle number concentration. It can be calculated by the following equation (Cai et al., 2018):*

$$N_{CCN}(SS) = \int_0^\infty AR(SS, D_P) N_{CN}(D_P) dD_p \qquad (4)$$

*where $N_{CCN}(SS)$ is the CCN concentration at a specific SS, $AR(SS, D_p)$ is the ratio of $N_{CCN}$ at a specific SS to $N_{CN}$ on a specific diameter from the SMCA method and $N_{CN}(D_P)$ is the particle number concentration at a specific diameter ($D_p$). Due to the absence of direct measurements for total $N_{CCN}$, we refer to the $N_{CCN}$ derived from Eq. (4) as observed values ($N_{CCN,obs}$) in this study. Previous research has shown that this method (size-resolved CCN from one column in CCNc-200) provides results closely matching those obtained from direct measurement (from another column in CCNc-200), supporting its reliability (Meng et al., 2014; Lathem and Nenes, 2011).*

*The $N_{CCN}$ (referred as $N_{CCN,sim}(SS)$) can be predicted by $D_{50}$ from closure method and $N_{CN}$ according to following equation:*

$$N_{CCN,sim}(SS) = \int_{D_{50,sim}(SS)}^\infty N_{CN}(D_P) dD_p \qquad (5)$$

*where the $D_{50,sim}(SS)$ was calculated based on the eq. (2) and (3).*

Reference:

Lathem, T. L. and Nenes, A.: Water Vapor Depletion in the DMT Continuous-Flow CCN Chamber: Effects on Supersaturation and Droplet Growth, Aerosol Sci Tech, 45, 604-615, doi: https://doi.org/10.1080/02786826.2010.551146, 2011.

Meng, J. W., Yeung, M. C., Li, Y. J., Lee, B. Y. L., and Chan, C. K.: Size-resolved cloud condensation nuclei (CCN) activity and closure analysis at the HKUST Supersite in Hong Kong, Atmos. Chem. Phys., 14, 10267-10282, doi: https://doi.org/10.5194/acp-14-10267-2014, 2014.

3.   Section 2.2.4: Regarding the cluster analysis, which method is used to classify the clusters? Is it a hierarchical clustering method? If so, is it bottom-up approach or top-down approach? If not, is k-means clustering method or fuzzy c-mean clustering method used? The authors should describe the method clearly. It is implied that the authors know exactly on what day the summer monsoon started. Can this be so clearly known? If so, explain clearly how so by showing the supporting data (e.g., wind pattern change, ⋯). Unlike ground (fixed location) measurement, cluster analysis for ship measurement requires some caution since the research vessel is moving (i.e., cruising). To ensure the representativeness and suitability of the midpoint used as the starting location, the back trajectories for the ship's coordinate during the cruises and the back trajectories for the midpoint of the ship track at the same time should be close enough. The authors should confirm if this is the case by showing supporting data.

Reply: Thanks for reviewer's valuable comment. The cluster analysis was performed by TrajStat, a plug-in module of MeteoInfo, based on k-means method (http://meteothink.org/docs/trajstat/cluster_cal.html). We have described the method in the manuscript (Lines 261-263):

*To clarify the sources of air masses, the cluster analysis was applied in this study, which was performed by TrajStat, a plug-in module of MeteoInfo, based on k-means method (http://meteothink.org/docs/trajstat/cluster_cal.html).*

As reported by the China Meteorological Administration (Chao et al., 2022), the summer monsoon in 2021 began during the sixth pentad of May. Based on the timing of the monsoon onset and the ship's actual trajectory, we selected two representative midpoints for the backward trajectory calculations. We have updated the wording throughout our manuscript for clarity and accuracy (Line 263-267):

*According to the report by the China Meteorological Administration (Chao et al., 2022), the summer monsoon in 2021 broke out during the sixth pentad of May. Therefore, based on the timing of the monsoon onset and the actual trajectory of the ship, we selected two representative midpoints of the ship track for backward trajectory calculations and cluster analysis in summer*

To ensure the representativeness and suitability of the midpoint used as the starting location, we compared the backward trajectories in midpoint and actual ship location. There was a slight difference between these two type back trajectories, indicating using the midpoint as the starting location could well represent the cluster analysis for ship measurement. We have shown the result in Fig. S8.

[Figure]

*Figure. S8. The backward trajectories of two midpoints (yellow and red line) and the location of research vessel (black line) during summer cruise (a) and winter cruise (b). The time interval for backward trajectories was 12 hours during the summer. Due to the shorter duration of the winter cruise, the time interval for the winter backward trajectories was set to 6 hours to more accurately distinguish the trajectory sources between the midpoint and real location.*


for 0.4% SS (0.74 vs. ~0.60). In the text, it is 0.72. A good example of poor sincerity of this manuscript! At 0.2% SS, summer and winter $\kappa$ were 0.49 and 0.31, respectively. Can they be considered "similar" as the authors stated? I do not think so. It should also be noted that the estimated $\kappa$ values are for the particles of critical diameter (D50), the smallest particles that can be activated at a given SS. So, these $\kappa$ values do not represent the $\kappa$ values of all the particles that can be activated at a given SS. This should be stated clearly before any arguments are made on $\kappa$ values.

Reply: We appreciate the reviewer for this valuable suggestion. We agree that using ratio of $N_{CCN}$ is more appropriate in this section. As suggested, we have replaced the absolute difference in $N_{CCN}$ between the two seasons with the ratio of $N_{CCN,winter}/N_{CCN,summer}$ at a given SS (Lines 323-326):

*The ratio of $N_{CCN}$ between summer and winter was smaller at high SS ($N_{CCN,winter}/N_{CCN,summer}$ = 0.51 and 0.54 at 0.4% SS and 0.7% SS, respectively) compared to low SS ($N_{CCN,winter}/N_{CCN,summer}$ = 0.62 at 0.2% SS), likely due to the significant difference in number concentration of Aitken-mode particles between the two seasons (Fig. 3a-b).*

We apologize for the oversight. We have rechecked the data in the tables and figures throughout the manuscript to ensure their accuracy. We have revised the $\kappa$ values (0.47 at 0.20% SS and 0.54 at 0.40% SS) in the text (line 330-333) and Table 1:

*Besides, the hygroscopicity pattern varied between seasons: in summer, $\kappa$ increased with SS (from 0.47 to 0.54 between 0.2% SS and 0.4% SS), while in winter, $\kappa$ decreased with SS (from 0.50 to 0.15 between 0.1% SS and 0.7% SS) (Fig. 3a-b).*

Additionally, we have revised the relevant statement (Line 330):
*The aerosol hygroscopicity ($\kappa$) was higher in summer than that in winter (Table 1).*

Moreover, we have acknowledged the limitation of $\kappa$ in Lines 205-206:
*Additionally, it is noting that the estimated $\kappa$ values refer to particles with the $D_{50}$, which are the smallest particles that can be activated at a given supersaturation.*

5.Section 3.2: To ensure that cluster analysis is well-conducted, I would suggest that the authors present all the back trajectories classified in each cluster in the supplement. If a back trajectory does not stay long enough within the boundary layer, it is difficult to say that it reflects the characteristics of the air masses where it passed. Therefore, it is recommended that the altitude of the back trajectory is also presented, to more clearly demonstrate the influence of the specified regions like "Mainland China", "Luzon", and "Indochinese Peninsula" on SCS. This can be confirmed by averaging the altitudes of all backward trajectories in each cluster. The authors state that low hygroscopicity of 'Mainland China' could be due to low sulfate concentration oxidized by DMS in winter than in summer. Here the comparison is between 'Mainland China' and 'Luzon,' the two terrestrial regions. So, I do not understand why DMS production is discussed here, which are definitely the source of CCN over marine regions. Explain the relevance.

Reply: Thank you for the reviewer's suggestion. We have incorporated the information on backward trajectories and clustering in Fig. S9, along with the average altitudes of each cluster presented in Fig. S10 (Lines 274-282).

*We further examined the trajectories for each cluster to verify their alignment with the air mass origins they represent (Fig. S9). The results demonstrate that cluster analysis*

*was well-conducted. Additionally, figure S10 illustrates the average altitude variation as the age in hours increases across different periods. During summer, the altitude of the clusters remained below 880 hPa, indicating that they resided within the boundary layer (about 800 hPa). While in winter, the altitude of the clusters was higher than in summer, especially for the cluster during the mixed period (peaked at about 755 hPa). However, these clusters were generally within or close to the boundary layer. These results suggest that the back trajectories could represent the characteristics of the air masses originating from these specified regions.*

[Figure]

*Figure S9. The backward trajectories of different clusters in summer (a)-(c) and winter (d)-(f).*

[Figure]

*Figure S10. The average pressure variation as age hour increased in different clusters in summer (a) and winter (b).*

We acknowledge that discussing the impact of DMS when comparing two periods affected by terrestrial air masses may not be suitable. We have revised the corresponding sentences and discuss the impact of DMS oxidation between summer and winter. In general, smaller particles generally exhibit lower hygroscopicity in winter (including the 'Mainland China' period) compared to summer, we speculated that sulfate from DMS oxidation likely influences the hygroscopicity of these particles. We have revised our statement accordingly (Lines 387-392):

*Additionally, hygroscopicity at smaller sizes was consistently lower across all winter periods, including the "Mainland China" period, compared to summer. This phenomenon may be related to the reduced sulfate fraction in smaller sizes during winter, as sulfate production via DMS oxidation is diminished due to lower sea surface temperatures in winter (18.0℃) compared to summer (29.3℃), which in turn inhibits DMS production by phytoplankton (Bates et al., 1987).*

Minor Comments

1. Line 117: "different impacts on CCN activity differently across seasons" --> "different impacts on CCN activity across the seasons"; What does "high cloud fraction" mean? Fraction of 'high cloud' or high fraction of clouds? If the former is the case, why is this relevant?

Reply: We have revised "high cloud fraction" to "fraction of high cloud." We would like to emphasize the seasonal variations in cloud properties over the SCS region, suggesting that aerosol-cloud interactions may differ across seasons. Therefore, it is crucial to conduct field campaigns on CCN activity throughout different seasons in the SCS region.

2. Line 134: "different monsoons" --> "summer and winter monsoons"

Reply: We have revised this expression.

3. Line 148: Add the information of the actual height of the sampling lines from the sea surface.

Reply: We have included information on the actual height of the sampling lines above the sea surface (Lines 148-150):

*On both cruises, most of the instruments were housed in a single compartment and the sampling lines were extended from the window of the compartment to the height of the ship's bridge (~17 m above sea level) (Fig. 1a).*

4. Lines 172-175: Since no result on OC/EC were discussed in the manuscript, it is inappropriate to mention OC/EC in Section 2.1.3. Likewise, the discussion of trace gases seems not to be presented in the manuscript and/or supplement. It might be worth checking.

Reply: We have removed the description of OC/EC and gas measurement.

5. Lines 181-182: Writing December 22nd as 12.22 can be confusing to readers. Unify the expression for date throughout the manuscript. Better to be December 22nd or 22 December but not 12.22.

Reply: We have changed "12.22" to "December 22$^{nd}$.

6. Line 186: 'praticle' --> 'particle'

Reply: It has been revised.

7. Line 191: "AR is the size-resolved AR" --> "AR indicates the size-resolved AR value" In several occasions later in the manuscript, AR seems to indicate the bulk AR value. These should be clearly explained.

Reply: We have revised this sentence. In the updated manuscript, we calculated the size-resolved AR through SMCA method and provided an explanation. AR has been defined throughout the manuscript, with 'bulk AR' referring to the bulk activation ratio and 'size-resolved AR' referring to the size-specific activation ratio."

8. Line 210: "κ from ..." --> "κ for ...", 'Nacl' --> 'NaCl' here and for other occasions.

Reply: We have corrected these two words.

9. Line 211: κ of organic was assumed to be 0.1. Where is it from?

Reply: We have added the reference in the end of the sentence (Lines 218-219):

*Besides, the κ of organic was 0.1 at this study according to Huang et al. (2022).*

10. Line 217: Eq. (4) can give predicted CCN concentration under the assumption that all particles of diameter greater than D50 activate for the given SS. Where is the justification?

Reply: We have added the reference in the end of this sentence (Lines 233-234):

*The $N_{CCN}$ can be predicted by $D_{50}$ from closure method and $N_{CN}$ according to following equation (Jurányi et al., 2011):*

11. Line 219: "number concentration under specific" --> "number density of specific"

Reply: We have revised this sentence.

12. Line 230-233: What does "have identical concentration at each size" mean? Is this intended for "fixed proportion for all sizes?" How does D50 calculated for each species? Explain clearly.

Reply: Yes, this indicates that the proportion of each component is independent of particle sizes. To avoid any confusion, we have revised the phrase "have identical concentration at each size" to "fixed proportion and hygroscopicity for all sizes". Additionally, we have explained the method for calculating D50 in the revised manuscript (Lines 246-250):

*(2) External-mixed scheme: the aerosol composition from the ToF-ACSM was assumed to be size-independent and externally mixed. Four type of aerosol (($NH_4)_2SO_4$, $NH_4NO_3$, NaCl and organic) are assumed to have a same proportion for all sizes. The $D_{50}$ from each species was calculated according to their κ values mentioned in 2.2.2. $N_{CCN}$ is calculated according to the Eq. (5) (Fig. S7b).*

We also added a figure in supplement to introduce these two schemes (Fig. S7).

[Figure]

*Figure S7. The internal and external simulation scheme*

13. Line 236: In Eq. (6), $N_{CCN,obs}$ is not an observed value, strictly speaking. Be clear on this.

Reply: Due to the malfunction of the column B, we calculated the total CCN concentration by combining the observed size-resolved AR with the actual particulate concentrations. This method has been validated in previous studies through comparisons with directly measured CCN concentrations, demonstrating close agreement and confirming its reliability (Meng et al., 2014; Lathem and Nenes, 2011). Therefore, in this study, we refer to the CCN concentration obtained through this method as $N_{CCN, obs}$. We have included an explanation in the manuscript to clarify why we designate it as $N_{CCN,obs}$   (Lines 222-232):

*Due to the malfunction of the column B, the CCN concentration (NCCN) was calculated based on size-resolved AR at a specific SS from SMCA method and observed particle number concentration. It can be calculated by the following equation (Cai et al., 2018):*

$$N_{CCN}(SS) = \int_0^\infty AR(SS,\, D_P)N_{CN}(D_P)dD_p \text{ (4)}$$

*where $N_{CCN}$ (SS) is the CCN concentration at a specific SS, AR(SS, $D_p$) is the ratio of $N_{CCN}$ at a specific SS to $N_{CN}$ on a specific diameter from the SMCA method and $N_{CN}(D_P)$ is the particle number concentration at a specific diameter (Dp). Due to the absence of direct measurements for total $N_{CCN}$, we refer to the $N_{CCN}$ derived from Eq. (4) as observed values ($N_{CCN,obs}$) in this study. Previous research has shown that this method (size-resolved CCN from one column in CCNc-200) provides results closely matching those obtained from direct measurement (from another column in CCNc-200), supporting its reliability (Meng et al., 2014; Lathem and Nenes, 2011).*

14. Line 246: 'outbreak' --> 'onset'

Reply: We have changed this word.

15. Lines 280-281: Aerosol number concentration is higher in summer than in winter, but mass concentration is higher in winter. Based on Figure 3, however, Aitken mode

particles are much more abundant in summer, while accumulation mode particles, which greatly affect mass concentration, are similar between summer and winter. Then, why are arosol mass concentration significantly different between summer and winter? Need more explanation.

Reply: Although the number concentration of accumulation mode appears similar between summer and winter, more particles in winter were concentrated in large sizes compared to those in summer, as shown by the particle volume size distribution (Fig. S15). And we explained it in the revision (Lines 318-321):

*Although $N_{CN}$ were higher in summer than in winter, the particle volume size distribution indicates that a higher fraction of particles was concentrated in larger size in winter, which significantly influenced mass concentration, resulting in a higher NR-$PM_1$ concentration (Fig. S15).*

[Figure]

*Figure S15. The average particle volume size distribution during summer and winter.*

16. Line 296: Add (Cai et al., 2020) after "Guangzhou", just like in the previous sentence.
Reply: We have added the reference after "Guangzhou".

17. Line 363: "NMB always lower" --> "NMB was always lower"
Reply: We have revised this sentence.

18. Line 386: "study size-resolved" --> "study of size-resolved"
Reply: We have added "of" in this sentence.

19. Line 414-415: "higher particle fraction in the accumulation mode compared to" --> "higher fraction of the accumulation mode particles comparted to"
Reply: We have revised this sentence.

20. Line 434: In "northern SCS," what does 'northern' exactly mean? The winter cruise route shown in Fig. 1 does not seem to indicate cruising over northern part of SCS. In

the same context, what does "remote SCS" mean?

Reply: Thanks for the reviewer's comment. We adopted definitions from existing literatures to categorize the South China Sea (SCS). The region near the Chinese mainland is referred to as the northern SCS, while the area farther from the mainland, near the Palawan Islands in the Philippines and Malaysia, is classified as the remote SCS (Atwood et al., 2017; Liang et al., 2021; Zhu et al., 2012).

We have provided additional explanations in the manuscript (Lines 144-148):
*Unfortunately, due to adverse weather conditions, such as strong winter monsoon winds causing poor sea conditions, and the fact that it was the first scientific deployment of the research vessel Sun Yat-sen University, the winter cruise had a shorter duration and covered a narrower spatial range, remaining only in the northern SCS (Fig. S1), compared to the summer cruise.*

[Figure]

*Figure S1. The definition of South China Sea from U.S. Energy Information Administration (https://www.eia.gov/international/analysis/regions-of-interest/South_China_Sea) and the yellow dash line and text were described the definition of northern and remote South China Sea according to other researches (Atwood et al., 2017; Liang et al., 2021; Zhu et al., 2012) (a); The cruises of these study (b).*

Reply: We have included information on $D_{50}$ from this study. In addition, we have clarified in the table that the AR refers to bulk AR.

*Table 1. The number concentration of particle and cloud condensation nuclei at different supersaturation (SS), the hygroscopicity and bulk activation ratio (AR), and activation diameter ($D_{50}$) at different SS in different studies.*

The winter cruise was conducted only in the northern SCS, and the specific definition of the northern South China Sea in this study is provided in Fig. S1.

Finally, we compared the result in Yellow Sea and our study in Line 326-329:

[revised manuscript text omitted]

23. Figure 1: Add important place names and mark the mid-points of back trajectories in (a). Provide full explanation in the caption.

Reply: We have added the important place names and mark the mid-points of back trajectories in Fig.1.

[Figure]

*Figure 1. The cruises of two shipborne observations, and the location of sample line and chimney of Tan Kah Kee, and Sun Yat-sen scientific vessel (a); Wind rose of the wind direction and wind speed in summer and winter cruises; The radius represents the frequency of wind direction occurrences, and the shaded areas indicate wind speed (b) and (c). The red circles are the midpoints of the ship trajectory selected for backward trajectory and cluster analysis in summer and the orange squares are the midpoints of the ship trajectory selected for backward trajectory and cluster analysis in winter.*

24. Figure 4: Show important place names.
Reply: We have added the important place names in Fig.4.

[Figure]

25. Figure 7: Why are there no κ plots for Indochinese Peninsula and Marine for 0.7% SS? Explain in the main text.
Reply: During these periods, the number concentration of particles smaller than 50 nm was relatively low (lower than 10 cm$^{-3}$). For the 0.7% SS settings, the D$_{50}$ fell within a small diameter range with low particle concentration, which would increase uncertainty in the corresponding κ values. We have clarified the reason for presenting only the κ value at 0.7% SS during the "Luzon" period in the main text. Additionally, we included Fig. S6 to illustrate the specific situation where we did not consider the fitting results at 0.7% supersaturation (Lines 207–210).

*During part of the summer measurement period, the $D_{50}$ at 0.7% supersaturation ranged between 30 and 40 nm. However, due to lower concentrations during these times, instrument noise introduced greater measurement uncertainty, as demonstrated in Fig. S6. Consequently, the average $D_{50}$ and $\kappa$ at 0.7% SS are not included in Table 1.*

[Figure]

*Figure S6. An example of $N_{CN}$ and $N_{CCN}$ timeseries during the Indochinese Peninsula (a) and Luzon periods (b).*

We also added the reason why we only presented data during "Luzon" period in Lines 352-356:

*Notably, we were able to obtain an accurate $D_{50}$ at 0.7% supersaturation only during the "Luzon" period in summer. Due to the relatively lower hygroscopicity compared to other summer periods, the corresponding $D_{50}$ at 0.7% SS ranged between 40 and 60 nm, with relatively high concentration of CN and CCN (Fig. S6), allowing for a more precise measurement of $D_{50}$. As a result, the $\kappa$ at 0.7% SS shown in Fig. 7 was specific to the Luzon period in summer.*

---

## Author Response (AR3)

We sincerely appreciate the reviewers for taking the time to review our manuscript and for providing valuable comments. Based on the reviewers' suggestions, we have revised the paper. Below are our responses to each of the reviewers' comments, with the reviewers' comments in black, our responses in red, and the revised manuscript content in italicized orange font.

**Reviewer 1**

The authors considered the review comments by the previous reviewers in revising the manuscript. The manuscript is written reasonably well as a measurement report. I have a few concerns that will need to be addressed before the manuscript can be considered for publication.

We sincerely thank the reviewer for their valuable feedback, which has helped us improve the quality of our manuscript. In response to the reviewer's comments, we have added details regarding ACSM calibration, compared the data from ACSM and SMPS, and elaborated on the importance and novelty of the closure method.

The ACSM calibration

The authors mentioned that they calibrated the instrument for RIE. However, there is no descriptions about the IE calibration. Regarding the CE, it is typically needed to compare the AMS/ACSM data with the aerosol mass concentration for checking the validity of CE. The authors at least have the SMPS data. I wonder how the authors validate the estimated values of CE.

Reply: Thanks for the reviewer's valuable comment. We calibrated the ionization efficiency (IE) value of $NO_3^-$ with the relative ionization efficiency (RIE) together. The calibration gives an IE value of 103.4 ions $pg^{-1}$ and 98.9 ions $pg^{-1}$ for nitrate in summer and winter cruises, respectively. We have included the information in lines 184-187:

*The ionization efficiency (IE) and relative ionization efficiency (RIE) values of the instrument were calibrated using ammonium nitrate ($NH_4NO_3$) and ammonium sulfate (($NH_4)_2SO_4$) both before the start and after the completion of the campaigns. The calibration gives an IE value of 103.4 ions $pg^{-1}$ and 98.9 ions $pg^{-1}$ for nitrate in summer and winter cruises, respectively.*

In addition, we compared the ACSM data with SMPS measurements to verify the CE value. An average particle density of 1.5 g $cm^{-3}$ was assumed to convert the PNSD data obtained from the SMPS into mass concentrations (Geller et al., 2006). Overall, the mass concentration time series measured by the ACSM and SMPS showed strong correlations, with correlation coefficients of 0.84 and 0.93 for summer and winter, respectively.

[Figure]

Figure S3. Comparison of mass concentration from ACSM and SMPS (a), the timeseries of mass concentration of ACSM and SMPS in summer (b), and the timeseries of mass concentration of ACSM and SMPS in winter (c).

However, before May 27 (prior to the onset of the summer monsoon), when air masses predominantly originating from Luzon in the Philippines were observed, SMPS-derived values consistently exceeded those measured by the ACSM. According to Chao et al. (2022), the summer monsoon onset occurred during the sixth pentad of May, which was approximately represented as May 27 for simplicity here. This discrepancy may be attributed to the ACSM's inability to detect certain refractory materials.

To further investigate this discrepancy, we compared black carbon concentrations during two distinct periods, utilizing measurements from the Aethalometer (Model AE33, Magee Scientific, USA). The differences in BC concentration between these periods were minor (0.67 $\mu$g m$^{-3}$ vs 0.48 $\mu$g m$^{-3}$), insufficient to account for the observed discrepancy between the SMPS-derived mass concentration and ACSM mass concentration. It is noteworthy that the AE33 might underestimate BC concentrations during May 5 to 27, owing to the lower detection efficiency for smaller black carbon particles (< 200 nm) relative to larger ones (Nakayama et al., 2010; Drinovec et al., 2015). Prior to May 27, the South China Sea region was predominantly influenced by air masses originating from Luzon. The particle size distribution centered a size range of 50-150 nm (Fig. 2a1 in the manuscript), aligning with the particle size distribution of black carbon from urban emissions reported in Schwarz et al. (2008). It implies that the black carbon might distribute at a relatively small particle size range, which could not fully be detected by the AE33, potentially contributing to the discrepancy between the SMPS-derived and ACSM-measured mass concentrations.

Additionally, we analyzed data from another campaign conducted over the South China Sea in June 2022. During this campaign, a typhoon (Chaba) altered local circulation patterns, leading to the transport of substantial pollutants from the Indochinese Peninsula to the ocean after June 28th (Fig. 1.1). Under these conditions, the mass concentrations measured by the SMPS were again consistently higher than those measured by the ACSM (Fig. 1.2), suggesting that the small size black carbon particle could be the primary factor underlying the mass discrepancy.

[Figure]

Figure 1.1. Timeseries of particle number size distribution in June 2022 in South China Sea.

[Figure]

Figure 1.2. Comparison of mass concentration from ACSM and SMPS (a), the timeseries of mass concentration of ACSM and SMPS (b).

A review of the literature indicates that discrepancies between SMPS and AMS/ACSM measurements have been observed at other locations as well (Sun et al., 2016; Wang et al., 2016; Kuang et al., 2020). When a CE of 0.5 was applied, the correlation coefficient for summer slightly increased from 0.84 to 0.85, though the overall difference remained negligible. Additionally, differences in measurement ranges and methodologies between the SMPS and ACSM are likely contributing factors to these discrepancies. We have added the relevant information in lines 191-193 of the manuscript:

*Detailed CE calculation and discussion can be found in the supplementary (Text S1, and Fig. S3). Assuming an average aerosol density of 1.5 g cm$^{-3}$ (Geller et al., 2006), the mass concentrations measured by the SMPS and ACSM exhibit a strong overall correlation, with correlation coefficients of 0.84 in summer and 0.93 in winter.*

We also have added the discussion in Text S1:

*In addition, the SMPS data was used to compared with ACSM data in order to verify the CE value. An average particle density of 1.5 g cm$^{-3}$ was assumed to convert the PNSD data obtained from the SMPS into mass concentrations (Geller et al., 2006). Overall, the mass concentration time series measured by the ACSM and SMPS showed strong correlations, with correlation coefficients of 0.84 and 0.93 for summer and winter, respectively. However, before May 27 (prior to the onset of the*

*summer monsoon), when air masses predominantly originating from Luzon in the Philippines were observed, SMPS-derived values consistently exceeded those measured by the ACSM. According to Chao et al. (2022), the summer monsoon onset occurred during the sixth pentad of May, which was approximately represented as May 27 for simplicity here. This discrepancy may be attributed to the ACSM's inability to detect certain refractory materials.*

*To further investigate this discrepancy, we compared black carbon concentrations during two distinct periods, utilizing measurements from the Aethalometer (model AE33). The differences in BC concentration between these periods were minor (0.67 $\mu$g m-3 vs 0.48 $\mu$g m-3), insufficient to account for the observed discrepancy between the SMPS-derived mass concentration and ACSM mass concentration. It is noteworthy that the AE33 might underestimate BC concentrations during May 5 to 27, owing to the lower detection efficiency for smaller black carbon particles (< 200 nm) relative to larger ones (Nakayama et al., 2010; Drinovec et al., 2015). Prior to May 27, the South China Sea region was predominantly influenced by air masses originating from Luzon. The particle size distribution centered a size range of 50-150 nm (Fig. 2a1 in the manuscript), aligning with the particle size distribution of black carbon from urban emissions reported in Schwarz et al. (2008). It implies that the black carbon might distribute at a relatively small particle size range, which could not fully be detected by the AE33, potentially contributing to the discrepancy between the SMPS-derived and ACSM-measured mass concentrations.*

*A review of the literature indicates that discrepancies between SMPS and AMS/ACSM measurements have been observed at other locations as well (Sun et al., 2016; Wang et al., 2016; Kuang et al., 2020). When a CE of 0.5 was applied, the correlation coefficient for summer slightly increased from 0.84 to 0.85, though the overall difference remained negligible. Additionally, differences in measurement ranges and methodologies between the SMPS and ACSM are likely contributing factors to these discrepancies.*

References:

Drinovec, L., Močnik, G., Zotter, P., Prévôt, A.S.H., Ruckstuhl, C., Coz, E., Rupakheti, M., Sciare, J., Müller, T., Wiedensohler, A., Hansen, A.D.A., 2015. The "dual-spot" Aethalometer: an improved measurement of aerosol black carbon with real-time loading compensation. Atmos. Meas. Tech. 8, 1965-1979.

Geller, M., Biswas, S., Sioutas, C., 2006. Determination of Particle Effective Density in Urban Environments with a Differential Mobility Analyzer and Aerosol Particle Mass Analyzer. Aerosol Sci Tech 40, 709-723.

Kuang, Y., He, Y., Xu, W., Zhao, P., Cheng, Y., Zhao, G., Tao, J., Ma, N., Su, H., Zhang, Y., Sun, J., Cheng, P., Yang, W., Zhang, S., Wu, C., Sun, Y., Zhao, C., 2020. Distinct diurnal variation in organic aerosol hygroscopicity and its relationship with oxygenated organic aerosol. Atmos. Chem. Phys. 20, 865-880.

Nakayama, T., Kondo, Y., Moteki, N., Sahu, L.K., Kinase, T., Kita, K., Matsumi, Y., 2010. Size-dependent correction factors for absorption measurements using filter-based photometers: PSAP and COSMOS. Journal of Aerosol Science 41, 333-343.

Schwarz, J. P., Gao, R. S., Spackman, J. R., Watts, L. A., Thomson, D. S., Fahey, D. W., Ryerson, T. B., Peischl, J., Holloway, J. S., Trainer, M., Frost, G. J., Baynard, T., Lack, D. A., de Gouw, J. A., Warneke, C., and Del Negro, L. A.: Measurement of the mixing state, mass, and optical size of individual black carbon particles in urban and biomass burning emissions, 35, doi:https://doi.org/https://doi.org/10.1029/2008GL033968, 2008.

Sun, Y., Jiang, Q., Xu, Y., Ma, Y., Zhang, Y., Liu, X., Li, W., Wang, F., Li, J., Wang, P., Li, Z., 2016. Aerosol characterization over the North China Plain: Haze life cycle and biomass burning impacts in summer.  121, 2508-2521.

Wang, Q., Zhao, J., Du, W., Ana, G., Wang, Z., Sun, L., Wang, Y., Zhang, F., Li, Z., Ye, X., Sun, Y., 2016. Characterization of submicron aerosols at a suburban site in central China. Atmos Environ. 131, 115-123.

Closure study

Such CCN closure studies have long been conducted. As the authors admit in the manuscript, the ACSM is unable to measure NaCl, leading to some uncertainties in the closure study. I could not understand the reason why calculations for the external mixing case were needed, even if the authors did not show any supporting data about mixing state of aerosol particles.

This manuscript is a measurement report. I personally did not feel that the addition of the closure study enhanced the quality of the manuscript from the perspective of reporting the data. It may make sense to consider deleting the section. If the authors think that the description is critical for the manuscript, it would be needed to add some additional descriptions about the importance and novelty of the closure study.

Reply: Thank you for the reviewer's suggestion. In the previous version, we conducted sensitivity experiments by cross-calculating cloud condensation nuclei (CCN) concentrations using the average $D_{50}$ and PNSD from different seasons to explore which factor—hygroscopicity or PNSD—primarily influences CCN concentrations across seasons. Based on previous reviewer comments and considering that this is a measurement report, we determined that applying the CCN closure method is a more standardized and appropriate approach for brief data analysis, as it effectively links chemical composition with CCN activity. Therefore, we tried to use the CCN closure method for a straightforward interpretation of our data. Although the ACSM does not fully capture the mass concentration of sea salt, its contribution to the overall aerosol mass fraction is not significant (Wu et al., 2022) and has a minor effect on our conclusions. Results from the CCN closure method indicate that summer aerosol and large-sized aerosol in winter predominantly existed in an internally mixed state, while small-sized aerosol in winter was primarily externally mixed. This external mixing state may partly explain the lower hygroscopicity of small-sized particles observed during winter. Thus, we believe that using the CCN closure method to interpret our data is a reasonable approach in this measurement report. We have added some additional descriptions about the importances and novelty of the closure study in lines 452-457:

*The CCN closure method is a widely used approach that connects CCN activity with aerosol chemical composition (Cai et al., 2018; Meng et al., 2014; Deng et al., 2013). Studies have demonstrated that the aerosol mixing state is crucial for accurately parameterizing CCN activity (Su et al., 2010; Wang et al., 2010; Ervens et al., 2010). Moreover, the CCN closure method provides a framework for investigating the influence of aerosol mixing states on CCN activity (Padró et al., 2012; Wang et al., 2018; Patel et al., 2021). In this study, we applied two schemes based on the CCN closure method, as described in Section 2.2.3, which consider aerosol composition and mixing state.*

References:

Wu, C.-H., Yuan, C.-S., Yen, P.-H., Yeh, M.-J., Soong, K.-Y., 2022. Diurnal and seasonal variation,

chemical characteristics, and source identification of marine fine particles at two remote islands in South China Sea: A superimposition effect of local emissions and long-range transport. Atmos Environ. 270, 118889.

We sincerely appreciate the reviewers for taking the time to review our manuscript and for providing valuable comments. Based on the reviewers' suggestions, we have revised the paper. Below are our responses to each of the reviewers' comments, with the reviewers' comments in black, our responses in red, and the revised manuscript content in italicized orange font.

**Reviewer 2**

This manuscript presents a comprehensive aerosol and CCN properties over the South China Sea (SCS) during summer and winter from two ship-based measurements. The aerosol size distribution, chemical composition of $PM_1$, hygroscopicity, and the CCN (with closure study) are examined and briefly analyzed, and the seasonal variations in the aerosol species and activation ratio is evident. This study provides extra data sources for future studies over the SCS. I think this manuscript holds the potential of publication, after considering and addressing the concerns and questions I have listed below.

We sincerely thank the reviewer for their valuable and insightful comments, which have significantly improved the quality of our manuscript. In response, we have included additional measurement-related information, addressed critical details that were previously omitted in certain sections, and revised and supplemented relevant figures and tables to enhance the clarity and coherence of our presentation.

**Measurements**

1. Please specify the instruments (CCN counter, SMPS, and DMA), measurable size range (and bin size if applicable), frequency, and instrumental uncertainty on the size-resolved number concentrations.

Reply: Thank you for the reviewer's valuable suggestions. The Scanning Mobility Particle Sizer (SMPS) comprises a Differential Mobility Analyzer (DMA) and a Condensation Particle Counter (CPC). In the Scanning Mobility CCN Analysis (SMCA) method, the SMPS is integrated with the CCN counter (CCNc). After particle size selection by the DMA, particles are directed simultaneously to both the CPC and the CCNc, so we did not need a separate DMA.

The DMA scanning ranges used in the SMPS during the two campaigns are specified in lines 160-161 of the manuscript: 10–500 nm for the summer campaign and 10–593 nm for the winter campaign. The CCNc operated with a scanning duration of 20 minutes for each supersaturation level, while the DMA completed a full particle size scanning cycle every 5 minutes. This additional detail has been included in lines 170-171 of the manuscript:

*During the measurement process, each supersaturation level was held constant for 20 minutes, with the DMA completing a full scanning cycle every 5 minutes.*

The uncertainty in the instrument's measurement of size-resolved particle number concentration is approximately 5%-6% (Morre et al. 2010) and we have included this information in lines 178-179:

*The uncertainty in the instrument's measurement of size-resolved particle number concentration is approximately 5%-6% (Morre et al. 2010).*

2. What is the supersaturation ramping time scale for the CCN column A?

Reply: Thanks for reviewer's question. During the measurements, supersaturation varied from 0.1% to 0.2%, 0.2% to 0.4%, and 0.4% to 0.7%, with temperature stabilization times ranging from a few seconds to tens of seconds. However, transitioning from 0.7% to 0.1% or 0.2% required approximately 5 minutes. In data processing, only cases where temperature remained stable during

the DMA scanning phase were selected. We have added the supersaturation ramping time scale on lines 171-176:

*During the measurement process, each supersaturation level was held constant for 20 minutes, with the DMA completing a full scanning cycle every 5 minutes. During the measurements, supersaturation levels varied incrementally between 0.1% and 0.2%, 0.2% and 0.4%, and 0.4% and 0.7%, with temperature stabilization times ranging from a few seconds to several tens of seconds. However, reducing the supersaturation from 0.7% to 0.1% or 0.2% required approximately 5 minutes for stabilization. For data processing, only instances where the temperature remained stable throughout the DMA scanning phase were included in the analysis.*

3. Also, the error or precision of the ACSM measured mass concentration should be specified after the composition-dependent collection efficiency correction.

Reply: Thanks for reviewer's comment. The values obtained using the time-independent CE method deviate by approximately 3% compared to those derived with a constant CE of 0.5. We have included this information in lines 191-193:

*The values obtained using the time-independent CE method show a deviation of approximately 3% compared to those obtained with a constant CE of 0.5.*

4. The uncertainties for the meteorological quantities need to be reported.

Reply: Thanks for reviewer's suggestion. The automatic weather stations on the Jiageng and Sun Yat-sen research vessels were manufactured by the Finnish company Vaisala, with model numbers AWS430 and WXT536, respectively. The AWS430 provides measurement accuracies of $\pm2\%$ for wind speed, $\pm2\%$ for wind direction, $\pm0.3°C$ for temperature, and $\pm1\%$ for relative humidity (within the range of 0–90%). Similarly, the WXT536 offers accuracies of $\pm3\%$ for wind speed, $\pm3\%$ for wind direction, $\pm0.3°C$ for temperature, and $\pm3\%$ for relative humidity (within the range of 0–90%) (http://www.vaisala.com). We have added their accuracy specifications in lines 202–210 of the revised manuscript:

*The meteorological elements, including temperature, relative humility (RH), wind speed, and wind direction, were measured by the combined automatic weather station (AWS430, Vaisala Inc., Finland) onboard the vessels (Sun et al., 2024). During the winter cruises, meteorology data before 12.22 was missed due to the calibration for the automatic weather station (WXT536, Vaisala Inc., Finland) before 12.22. The timeseries of meteorological data were presented in Fig. S4. The AWS430 provides measurement accuracies of $\pm2\%$ for wind speed, $\pm2\%$ for wind direction, $\pm0.3°C$ for temperature, and $\pm1\%$ for relative humidity (within the range of 0–90%). Similarly, the WXT536 offers accuracies of $\pm3\%$ for wind speed, $\pm3\%$ for wind direction, $\pm0.3°C$ for temperature, and $\pm3\%$ for relative humidity (within the range of 0–90%) ([www.vaisala.com](www.vaisala.com)).*

5. More details on the aethalometer are also encouraged (size ranges, uncertainty, etc.).

Reply: Thanks for reviewer's suggestion. The AE33 measures the black carbon (BC) mass concentration in $PM_{2.5}$. These concentrations are referred to as equivalent BC mass concentrations because they represent the light absorption of BC at a wavelength of 880 nm. We have incorporated the relevant details about the AE33 in lines 195–200 of the manuscript:

*The black carbon (BC) mass concentrations were measured using an aethalometer (Model AE33, Magee Scientific, USA) with a 1-minute time resolution (Drinovec et al., 2015). Notably, the BC*

*mass concentrations obtained from AE33 are referred to as equivalent BC mass concentrations, as they represent the combined light absorption of BC at 880 nm. Prior to entering the AE33, the sampled air was passed through a PM$_{2.5}$ cyclone (BGI Inc., Waltham, MA, USA) to exclude particles larger than 2.5 µm.*

6. In the CCN closure section, can you report representative D50 values for four species (as in S7) under the External scheme, during summer and winter seasons in the study domain?

Reply: We have reported the D50 calculated by Eq. (2) according to the hygroscopicity of different species in Table S1. We have included the information in lines 275-279:

*(2)  External-mixed scheme: the aerosol composition from the ToF-ACSM was assumed to be size-independent and externally mixed. Four type of aerosol ((NH$_4$)$_2$SO$_4$, NH$_4$NO$_3$, NaCl and organic) are assumed to have a same proportion for all sizes. The D50 from each species was calculated by Eq. (2) according to their κ values mentioned in 2.2.2. N$_{CCN}$ is calculated according to the Eq. (5) (Fig. S8b and Table S1).*

| D$_{50}$ (nm) | 0.1% SS | 0.2% SS | 0.4% SS | 0.7% SS |
|---|---|---|---|---|
| Sulfate | 143 | 90 | 57 | 39 |
| Seasalt | 109 | 69 | 43 | 30 |
| Nitrate | 135 | 85 | 53 | 37 |
| Organic | 242 | 192 | 152 | 126 |

**Table S1. The D$_{50}$ of different species in external scheme.**

I believe those are crucial for a measurement report.

Specific Comments:

Line 117: Can you elaborate on how the seasonal variation of the fraction of high cloud is relevant here in terms of the ship-observed CCN near the sea surface and the aerosol-high cloud interaction? Intuitively, would the lower boundary layer aerosol/CCN have greater impacts on the MBL clouds, or is there any particular dynamical mechanism the author is referring to?

Reply: Thanks for reviewer's comment. Given the absence of a well-established mechanism linking boundary layer aerosol properties and cloud condensation nuclei concentrations to the formation of high-altitude clouds, we revised the sentence to enhance clarity and cited new references that specifically discuss the influence of boundary layer aerosol variations on cumulus cloud properties over the South China Sea (Lines 119-122):

*Additionally, when the marine boundary layer over the SCS is influenced by various natural and anthropogenic sources, resulting in altered aerosol properties, the characteristics of cumulus clouds are correspondingly affected (Miller et al., 2023). This indicates that aerosol-cloud interactions vary between winter and summer seasons.*

L153. 'The ⋯ SMCA⋯ was initially⋯'. And do you want to say you utilize this sampling strategy to get the size-resolved CCN?

Reply: Thanks for the reviewer's valuable comment. We apologize for the lack of clarity in the original sentence. What we intended to convey is: The size-resolved CCN activity was measured using the scanning mobility CCN analysis (SMCA) method proposed by Moore et al. (2010), employing a combination of a scanning mobility particle sizer (SMPS) system and a cloud

condensation nuclei counter (CCNc-200, DMT Inc., USA). We have revised the sentence accordingly (Lines 155-157):

*The size-resolved CCN activity was measured using the scanning mobility CCN analysis (SMCA) method proposed by Moore et al. (2010), employing a combination of a scanning mobility particle sizer (SMPS) system and a cloud condensation nuclei counter (CCNc-200, DMT Inc., USA) (Fig. S2).*

L159. Have both sensors in the counter column B on two ships malfunctioned during the two sampling periods?

Reply: Thanks for reviewer's comment. Column B in CCNc was not used on both cruises. We have added the information in lines 161-162:

*Unfortunately, due to the malfunction of flow sensor in the column B on both cruises, only the data from column A is presented in this study.*

L206. You have introduced D50 as 'particle size at which 50% of the particles are activated at a specific SS' before. Your statement here is flawed.

Reply: Thank you for the reviewer's valuable feedback. We apologize for the error and have corrected this statement (Line 230):

*Additionally, it is noting that the estimated $\kappa$ values refer to particles with the $D_{50}$.*

L275. The results in S8 and S9 are well-justified, though, you should consider putting a few words about it in the main text.

Reply: Thanks for reviewer's comment. We have added some words about the Figure S8 (Lines 314-316):

*Minor discrepancies may exist between the air mass origins at certain midpoints and the actual ship locations. However, overall, the air mass origins at the midpoints are representative of those at the actual locations.*

L307. Winter period shows two peaks with more organic (less sulfate) and with both high organic and sulfate. Elaborate on how the impact of Northeast Monsoon 'persist'.

Reply: Thank you for the reviewer's comments. During the winter cruise, the winter monsoon was already dominant. The presence of two distinct peaks suggests that the two phases were likely influenced by air masses from different sources. As discussed later, the "Mainland China" phase exhibited a higher proportion of organic matter, while both sulfate and organic matter were relatively elevated during the "Mixed" phase.

L311. Define clearly how the Nucleation, Aikten, and Accumulation are defined, in terms of size cut and rationale, in this study (should have been done in the Data section).

Reply: Thanks for reviewer's comment. The Nucleation mode, Aikten, and Accumulation are defined by their geometric mean diameters (GMD). The GMD for nucleation modes (GMD1) typically ranges from 3 to 30 nm, for Aitken modes (GMD2) from 30 to 100 nm, and for accumulation modes (GMD3) above 100 nm. We have added a section in the Data section to describe the three modes (Section 2.2.4, lines 284-298):

*The multi-lognormal distribution function (Eq. (8)) is used to parameterize and optimize the descriptions of the measured PNSD (Heintzenberg, 1994) and is widely applied in aerosol research (Cai et al, 2020; Boyer et al., 2023; Zhu and Wang, 2024). An automatic mode-fitting algorithm (Hussein et al., 2005) is used to generate the model-fitted results.*

$$f\left(D_p, \bar{D}_{pg,i}, N_i, \sigma_{g,i}\right) = \sum_{i=1}^{n} \frac{N_i}{\sqrt{2\pi} \log(\sigma_{g,i})} \times exp\left[-\frac{\left[\log D_p - \log \bar{D}_{pg,i}\right]^2}{2\left(\log \sigma_{g,i}\right)^2}\right] \quad (8)$$

*where $D_P$ is the diameter of a particle. Each lognormal mode is characterized by three parameters: the mode number concentration ($N_i$), geometric variance ($\sigma_{g,i}$), and geometric mean diameter (GMD, $\bar{D}_{pg,i}$). The total number of lognormal modes used to describe the PNSD is denoted by n. These modes are fitted using an algorithm applied to each particle size distribution, with one to three lognormal distributions used per time step. The algorithm classifies the PNSD into nucleation, Aitken, and accumulation modes based on their geometric mean diameters (GMDs). The GMD for nucleation modes (GMD1) typically ranges from 3 to 30 nm, for Aitken modes (GMD2) from 30 to 100 nm, and for accumulation modes (GMD3) above 100 nm (Heintzenberg, 1994; Hussein et al., 2005; Zhu and Wang, 2024).*

L330. Explanation is needed on the flipped hygroscopicity-supersuration relations between summer and winter.

Reply: Thanks for reviewer's comment. We have explained the possible reason in lines 379-384:

*This contrasting trend may be related to the reduced sulfate fraction in smaller sizes during winter, as sulfate production via DMS oxidation is diminished due to lower sea surface temperatures in winter (18.0℃) compared to summer (29.3℃), which in turn inhibits DMS production by phytoplankton (Bates et al., 1987; Kouvarakis and Mihalopoulos, 2002). Additionally, it could be linked to the mixing state of the particles, with further discussion provided in the following sections.*

L388. Which figure or table are you referring to wrt. 'smaller sizes'. And yes, sulfate fraction is reduced in winter 'Marine' period, but the increased ammonium may compensate for this effect (Fig. 5f). You may also consider attributing this to the increases in organic aerosol contribution due to factors like reduced photochemical oxidation.

Reply: Thank you for the reviewer's comment. The smaller sizes refer to hygroscopicity at high supersaturation (0.7% SS), since the $D_{50}$ at high SS was much lower than those at low SS. We have included this information in lines 233-235:

*According to κ-Köhler theory, in the following discussion, the hygroscopicity of small particles is associated with hygroscopicity at high SS, whereas the hygroscopicity of large particles is linked to hygroscopicity at low SS.*

We modified the sentence in L388 (in previous version) to clarify:

*Additionally, as discussed in Section 3.1, the reduced biological activity during winter, which results in a decline in the fraction of small-particle sulfate and an increase in the fraction of organics, may also contribute to this low hygroscopicity in small particles (at high SS, fig 7b).*

We also consider that the higher fraction of organic aerosols at smaller sizes, resulting from the lower sulfate concentrations due to reduced biological activity in winter, may contribute to the lower hygroscopicity at high SS. We have added the relevant information in lines 437-441:

*This suggests that the lower hygroscopicity in smaller particles during the "Mainland China" period may be attributed to a larger fraction of hydrophobic BC. Additionally, as discussed in Section 3.1, the reduced biological activity during winter, which results in a decline in the fraction of small-particle sulfate and an increase in the fraction of organics, may also contribute to this low hygroscopicity in small particles.*

Technical comments:

L56. '…partially attributed…'. And considering adding more recent and relevant references to these statements.

Reply: We have corrected the word and added more references.

L71. Please be more specific on what particles (compositions, sizes, etc.) were examined in Ajith et al. (2022)

Reply: In Ajith et al. (2022), "particles" refer to total particles without specifying any particle type or size range. According to Köhler theory, only small particles are considered in discussions about their activation as cloud condensation nuclei (CCN), whereas larger particles can act as CCN directly.

L90. This paper (Zheng et al., 2020) is not on the reference list. And I presume you refer to 'Eastern North Atlantic'.

Reply: Thanks for reviewer's comments. We have added the reference and corrected the location to "Eastern North Atlantic".

L153. 'The … SMCA… was initially…'.

Reply: We have corrected the word.

L153. '(Fig. 1c2)'

Reply: We consider that we can reference Figure S2 here.

L377. Here, the statement, though reasonable, has not been supported by the results, use 'potentially led' instead.

Reply: We have corrected the word.

Figures. Please put the figure caption directly beneath the figure.

Reply: We have put the figure caption beneath the figure.

Fig. 3. Same y-axis range is needed for (a) and (b). And please state the seasons for (e) and (f).

Reply: We have redrawn Figure 3.

[Figure]

Figure 3. Particle number size distribution in summer (a) and winter (b); The red markers represent the activation diameters and hygroscopicity parameters corresponding to 0.1%, 0.2%, 0.4%, and 0.7% supersaturations in this study (without 0.1% in summer). The green markers represent the hygroscopicity parameters reported in Atwood et al. (2017) for the southern South China Sea during summer. The gray markers represent the hygroscopicity parameters documented in Cai et al. (2018) for the Pearl River Delta region during winter. The fraction of NR-PM1 in summer (c) and winter (d) in this study, in northern SCS reported by Liang et al. (2021) (e), and in North Pacific reported by Choi et al. (2017) (f).

Fig. S9. Panel (c) and (f), consider using something like 'Marine-Win'? I was confused with Marine-(South) and Marine-(West) at first glance.
Reply: We have redrawn the Figure S10 (Original Fig. S9).

[Figure]

Figure S10. The backward trajectories of different clusters in summer (a) and winter (b).

Fig. S14. Use more distinguishable colors between 14.6 and 41.4 nm. And the subpanel labels do not match the captions for winter.

Reply: We have redrawn the Figure 15 (Original Fig. S14):

**Summer**

[Figure]

**Winter**

Figure S15. Timeseries of particle number size distribution (a) and (e), mass concentration of NR-PM1 (b) and (f), particle number concentration in 14.6, 41.4, and 109.4 nm (c) and (g), mass concentration of black carbon (d) and (h); The figure number from (a) to (d) means the data in summer, and the figure nnmber from (e) to (h) means the data in winter; The number 1 represented the data before data quality control and the number 2 represent the data after data quality control.

Reference. The reference list is hard to distinguish between entries, so please correct the format.
Reply: We have changed the format on the reference list to make it clear.

---

## Author Response (AR4)

We sincerely appreciate the reviewer for taking the time to review our manuscript and for providing valuable feedback. Based on the reviewers' suggestions, we have revised the paper. Below, we provide our responses to each of the reviewers' comments. The reviewers' comments are presented in black, our responses in red, and the revised manuscript content is highlighted in italicized orange font.

I would like to thank the authors for their efforts in carefully considering and addressing the reviewers' comments, which have substantially improved the clarity of the manuscript. This study makes a significant and informative contribution to aerosol and CCN research in the SCS. I have only a few minor suggestions before it might be published.

Reply: We appreciate the reviewers' recognition of our work and are grateful for the valuable feedback provided by the reviewers and editors. In this revised version, we have made changes based on the reviewers' suggestions, added relevant information, and acknowledged the support and assistance of the reviewers and editors in the Acknowledgment section.

The discussion in the authors' responses regarding the analysis of another campaign's results on the ACSM/SMPS comparison— '…we analyzed data from another campaign conducted over the South China Sea in June 2022…'—provides valuable insights into the mass discrepancy. This part, along with the associated figures, should also be included in the supplementary material.

Reply: Thank you for the reviewer's suggestion. We have discussed the discrepancies between the ACSM and SMPS during the 2022 campaign in the Supplement (Supplement, Lines 67-72).

*Additionally, we analyzed data from another campaign conducted over the South China Sea in June 2022. During this campaign, a typhoon (Chaba) altered local circulation patterns, leading to the transport of substantial pollutants from the Indochinese Peninsula to the ocean after June 28th (Fig. S19). Under these conditions, the mass concentratio measured by the SMPS were again consistently higher than those measured by the ACSM (Fig. S20), suggesting that the small size black carbon particle could be the primary factor underlying the mass discrepancy.*

[Figure]

**Figure S1. Timeseries of particle number size distribution in June 2022 in South China Sea.**

[Figure]

**Figure S2. Comparison of mass concentration from ACSM and SMPS (a), the timeseries of mass concentration of ACSM and SMPS (b).**

In the main text, a brief mention of the potential or speculated causes of the ACSM and SMPS mass discrepancies should be added after the statement on Line 193: '…the mass concentrations measured by the SMPS and ACSM exhibit a strong overall correlation, with correlation coefficients of 0.84 in summer and 0.93 in winter…'.

Reply: Thank you for the reviewer's suggestion. We have added information about the discrepancies between the ACSM and SMPS during certain periods after the statement on Line 193 (Lines 195-197):

*During the pre-onset phase of the summer monsoon (prior to May 24), periodic discrepancies were observed between the ACSM and SMPS data, likely due to the influence of refractory aerosol (ie. Black carbon). This issue is discussed in detail in Text S1.*

Finally, I note that the manuscript has been significantly improved through multiple rounds of peer-reviews. The efforts of all the reviewers and the editor throughout this process should be acknowledged in the Acknowledgments section.

Reply: We thank the reviewer for the valuable comment. We deeply appreciate the constructive feedback provided by both the reviewers and the editors, which has greatly enhanced the quality of our manuscript. Acknowledgment of their contributions has been included in the Acknowledgment section:

*We sincerely thank the reviewers and editors for their valuable suggestions during the review process, which have been instrumental in enhancing the quality of this manuscript. Lastly, we wish to honor the memory of Professor Zhao Jun and express our heartfelt gratitude for his significant contributions to this work.*

---

## Author Response (AR5)

We sincerely thank the editor for the valuable suggestions, which have been carefully considered and incorporated into the revised manuscript. Below, we provide our responses to each of the editor's comments. The editor's comments are presented in black, our responses in red, and the revised manuscript content is highlighted in italicized orange font.

Dear Authors,

Thanks for attention to the final reviewer comments. I suspect the reference to a new Figure S2 in the responses was meant to be called Figure S4?

Reply: We apologize for the incorrect figure citation. This error occurred because the field codes from Word were inadvertently copied when transferring the figure captions from the Supplement to the response document, leading to the incorrect display of Figure S2. The figures have now been updated to Figure S19 and Figure S20 and are correctly cited in Text S1 (Supplement, Lines: 72-77):

*Additionally, we analyzed data from another campaign conducted over the South China Sea in June 2022. During this campaign, a typhoon (Chaba) altered local circulation patterns, leading to the transport of substantial pollutants from the Indochinese Peninsula to the ocean after June 28th (Fig. S19). Under these conditions, the mass concentrations measured by the SMPS were again consistently higher than those measured by the ACSM (Fig. S20), suggesting that the small size black carbon particle could be the primary factor underlying the mass discrepancy.*

[Figure]

**Figure S19. Timeseries of particle number size distribution in June 2022 in South China Sea.**

[Figure]

**Figure S20. Comparison of mass concentration from ACSM and SMPS (a), the timeseries of mass concentration of ACSM and SMPS (b).**

I request a final set of minor corrections, including telling readers how you derived a mass concentration explicitly in Figure S4 from the SMPS since the SMPS measures volume concentration.

Reply: Thank you for the editor's suggestion. We had previously provided a brief explanation in Text S1 on how to convert dN/dlogDp to mass concentration. To improve clarity, we have now added Equation 7 and Equation 8 in Text S1 along with a detailed explanation of the calculation process (Supplement, Lines: 45-52):

*In addition, the SMPS data was used to compared with ACSM data in order to verify the CE value. An average particle density ($\rho$) of 1.5 g cm$^{-3}$ was assumed to convert the PNSD data obtained from the SMPS into mass concentrations (Geller et al., 2006) according to Eq. (7) and Eq. (8):*

$$\frac{dV}{dlogDp} = \frac{\pi}{6} D_p^3 \frac{dN}{dlogDp} \tag{7}$$

$$M_{SMPS} = \int_{D_{p,min}}^{D_{p,max}} \rho \frac{dV}{dlogDp} dlogDp \tag{8}$$

*where $M_{SMPS}$ is the mass concentration from SMPS, $D_{p,min}$ and $D_{p,max}$ refer to the minimum and maximum particle sizes scanned by the SMPS. dN/dlogDp and dV/dlogDp are particle number size distribution and particle volume size distribution which could be measured by SMPS.*

Lastly, please check that you add units to all figure axes when missing such as for dN/dlogDp in Figure S15 and dV/dlogDp in Figure S16. There are additional typos that need to be cleaned up like in caption in Figure S16 (e.g., "volumn"). A final cleaning of both the article and SI file for such details would be helpful prior to advancing to publication.

Reply: Thank you for the valuable suggestion. We have carefully reviewed the figures (ie. Figure 2, Figure S13, Figure S15, and Figure S16) and their captions (ie. Figure S16) in the appendix to ensure that both axes are labeled with appropriate units and that the captions clearly and accurately describe the content of the figures.

[Figure]

**Figure 2.** Timeseries of (a) particle number size distribution, (b) mass concentration of NR-PM1, and (c) its fraction, (d) number concentration of total particle and cloud condensation nuclei under the supersaturation of 0.1%, 0.2%, 0.4%, and 0.7%, and (e) aerosol hygroscopicity. The number 1 in figure number means timeseries in summer and number 2 means it in winter.

[Figure]

**Figure S13. Wind rose of the relative wind direction (with respect to the bow) and relative wind speed (with respect to the ship speed) in summer and winter cruises; The radius represents the frequency of wind direction occurrences, and the shaded areas indicate wind speed (a1) and (b1); Wind rose of the wind direction and wind speed in summer and winter (a2) and (b2).**

**Summer**

[Figure]

**Winter**

**Figure S15.** Timeseries of particle number size distribution (a) and (e), mass concentration of NR-PM₁ (b) and (f), particle number concentration in 14.6, 41.4, and 109.4 nm (c) and (g), mass concentration of black carbon (d) and (h); The figure letters from (a) to (d) mean the data in summer, and the figure letters from (e) to (h) mean the data in winter; The number 1 represented the data before data quality control and the number 2 represent the data after data quality control.

[Figure]

**Figure S16. The average particle volume size distribution during summer and winter.**